# Complement is activated by elevated IgG3 hexameric platforms and deposits C4b onto distinct antibody domains

Leoni Abendstein [1], Douwe J. Dijkstra [2], Rayman T. N. Tjokrodirijo[3], Peter A. van Veelen [3], Leendert A. Trouw [2], Paul J. Hensbergen [3] & Thomas H. Sharp [1]

IgG3 is unique among the IgG subclasses due to its extended hinge, allotypic diversity and enhanced effector functions, including highly efficient pathogen neutralisation and complement activation. It is also underrepresented as an immunotherapeutic candidate, partly due to a lack of structural information. Here, we use cryoEM to solve structures of antigen-bound IgG3 alone and in complex with complement components. These structures reveal a propensity for IgG3-Fab clustering, which is possible due to the IgG3-specific flexible upper hinge region and may maximise pathogen neutralisation by forming high-density antibody arrays. IgG3 forms elevated hexameric Fc platforms that extend above the protein corona to maximise binding to receptors and the complement C1 complex, which here adopts a unique protease conformation that may precede C1 activation. Mass spectrometry reveals that C1 deposits C4b directly onto specific IgG3 residues proximal to the Fab domains. Structural analysis shows this to be caused by the height of the C1-IgG3 complex. Together, these data provide structural insights into the role of the unique IgG3 extended hinge, which will aid the development and design of upcoming immunotherapeutics based on IgG3.

Antibodies bind specifically and strongly to antigens on invading pathogens and mediate potent immune effector functions. Immunoglobulin G (IgG) is the predominant human immunoglobulin class in blood and comprises four subclasses that are named after their abundance in human serum; of all IgG in plasma, ~60% are IgG1, followed by ~30% IgG2, ~7% IgG3 and ~3% IgG4[1–3]. IgG subclasses are highly similar, sharing ~95% sequence homology in the heavy chain constant domains, and all contain a fragment crystallisable (Fc) domain linked to two fragment antigen-binding (Fab) domains via a disulfide-linked polyproline hinge[1–4]. IgG1, IgG2 and IgG4 are structurally homologous, with relatively short and rigid hinge regions containing 2-4 disulfide bonds, which limit the flexibility and size of the antibody complex[2,4–7]. In contrast, IgG3 contains a uniquely long hinge formed from a triple repeat of 15 amino acids that introduces 11 disulfide bonds and three O-linked glycosylation sites on each of the hinge polypeptides (Supplementary Fig. 1a)[1,2,8]. Due to this long hinge, IgG3 is the most flexible subclass, with longer Fc-Fab and Fab-Fab distances than other subclasses[6]. Furthermore, IgG3 has the greatest allotypic variation; ~15 allotypes have been discovered so far with various hinge lengths that, combined with differential O-glycosylation occupancy, can vary greatly in their effector functions and structures[1,8,9].

IgG3 is relatively less studied compared to other subclasses, in part due to its structural and functional heterogeneity, which has led to IgG3 being neglected as an immunotherapeutic candidate[2,3,10,11]. This is in stark contrast with data describing the critical role of IgG3 in protection against pathogens[2], with IgG3 having more potent effector functions

[1]Department of Cell and Chemical Biology, Leiden University Medical Center, 2300 RC Leiden, The Netherlands. [2]Department of Immunology, Leiden University Medical Center, 2333 ZA Leiden, The Netherlands. [3]Center for Proteomics and Metabolomics, Leiden University Medical Center, 2333 ZA Leiden, The Netherlands. ✉e-mail: t.sharp@lumc.nl

than other IgG subclasses regarding pathogen neutralisation[12,13], antibody-dependent cellular phagocytosis (ADCP)[12,14,15], intracellular antiviral immunity[16], and complement activation[1,2,5,8,17–19]. Recent advances in understanding IgG3 functions have revealed a critical role for protection against viruses, particularly SARS-CoV-2[13,20] and HIV[16,21], which can synergise with complement activation to both neutralise virions and inhibit their disassociation within cells[21].

The classical complement pathway is activated when the initiating complement component 1 (C1) complex, composed of C1q and a heterotetramer of $C1r_2s_2$, binds to antigen-associated antibodies (Supplementary Fig. 1b, c). Oligomerization of antigen-bound IgG presents C1q binding sites on a hexagonal platform comprising of a ring of Fc domains[22]. C1q binds these hexagonal Fc-platforms and activates the C1r serine protease (SP), which then cleaves and activates the C1s SP. C1s propagates the cascade by cleaving C4 to form C4b, which undergoes a large structural rearrangement that exposes a reactive thioester group and enables covalent conjugation of C4b onto exposed surfaces and molecules[23,24]. Complement progression results in opsonization of the target membrane and formation of the membrane attack complex (MAC) pore, which perforates the lipid bilayer and lyses the cell (Supplementary Fig. 1c)[25].

Here, we use a combination of biophysical assays, mass spectrometry and cryo-electron tomography to provide structural data of antigen-bound IgG3. We observe arrays of densely-spaced Fab domains forming on lipid membranes, which could maximise the potential for pathogen neutralisation. Above these ordered Fab domains, elevated hexameric Fc platforms are visible that are consistently far from the liposome membrane and more highly exposed than for IgG1, which may reduce steric barriers to receptor binding. Upon the addition of complement components, we observe C1 complexes exclusively binding to these elevated IgG3-Fc platforms, and C1 adopts configurations related to C1 autoactivation and C4b deposition. Subtomogram maps of IgG3-C1-C4b reveal the importance of the extended IgG3 hinge, as it allows deposition of C4b onto the IgG3 Fab-hinge interface, with significant consequences for complement activation. These structural insights provide a wealth of data for the development and design of upcoming immunotherapeutics based on IgG3, especially regarding the length of IgG hinges for a given target antigen.

## Results

### Antigen-bound IgG3 forms ordered Fab domains
Human anti-2,4-dinitrophenyl (DNP) IgG1 and IgG3 (allotype G3m5[1,2]) were purified and their ability to activate complement confirmed using enzyme-linked immunosorbent assays (ELISA), which consistently showed better C1q binding to antigen-bound IgG3 compared to IgG1, as well as more efficient C4 deposition and MAC pore formation (Supplementary Fig. 2). To assess the ability of membrane-associated antigen-bound IgG subclasses to activate complement, we used liposomes displaying 1 mol% DNP haptens and encapsulating a self-quenching concentration of the dye sulforhodamine B. Complement activation leads to MAC pore formation[25], which causes leakage of the dye and a corresponding increase in fluorescence[24,26,27]. Incubation of antigenic liposomes with IgG and 10% normal human serum (NHS) showed that IgG3 was better than IgG1 at performing complement-mediated membrane lysis (Supplementary Fig. 2c). To exclude the effect of differential glycosylation of IgG1 and IgG3 Fc domains affecting C1q binding[28], we assessed the Fc N-glycosylation profile on IgG1 and IgG3, respectively. Tryptic digests of purified IgG1 and IgG3 revealed comparable glycosylation (Supplementary Fig. 3), with the presence of identical N-linked complex type glycans on both subclasses at the highly-conserved N297 residue within the $C_H2$-domain (Supplementary Table 1).

Antigenic liposomes with 1 mol% DNP were used as cell mimetics for cryo-electron tomography[29]. Incubation of antigenic liposomes with IgG1 displayed abundant antibody binding (Fig. 1a and

Supplementary Fig. 4a). Antibodies were frequently found clustered into patches on the liposome surfaces, presumably due to membrane fluidity allowing antigen-bound IgG1 monomers to diffuse laterally until contacting another monomer. IgG1 oligomerization is driven by the formation of hexameric platforms, and in keeping with this, IgG1 platforms were detected with a height of ~11 nm above the lipid membrane (Fig. 1a and Supplementary Fig. 4a), as previously described[30]. Density corresponding to Fab domains was visible between the platforms and the lipid membrane, although there was no regular structure apparent.

In contrast, liposomes incubated with IgG3 revealed ordered Fab domains bound to the antigenic lipid membranes (Fig. 1b and Supplementary Fig. 4b). Striations perpendicular to the membrane were visible covering extensive patches on the liposome surfaces. These striations were measured to be ~7 nm high (Fig. 1b), and Fourier analysis gave a repeating width of 3.8 nm (Supplementary Fig. 5a, b). A low-resolution 2D average enabled measuring the distance between the striations, again yielding a separation of 3.8 nm (Fig. 1c, d). In some tomographic slices we observed small, rectangular patches of ordered Fab arrays in the imaging plane at the zenith of liposomes (Supplementary Fig. 5c). Alignment of these patches revealed ordered arrays of Fab domains (Fig. 1c–f), and Fourier analysis identified repeating structures at 7.6 nm and 3.8 nm in one direction, and 5.25 nm in a direction 76° offset (Supplementary Fig. 5d). The 7.6 nm peak is twice the 3.8 nm distance observed in Supplementary Fig. 5b, and corresponds to a pair of Fab domains, each 3.8 nm apart. These measured distances, 7 nm high and 3.8 × 5.25 nm wide, correspond precisely with the height and width of individual Fab domains (Supplementary Fig. 5e). We identified various Fab-Fab interfaces present in crystal structures in the protein database (PDB) present in multiple crystal structures. Through analysis of these interfaces, we identified several candidates that exhibited the repeat distances observed here (Supplementary Fig. 6). One of the models, with PDB code 5TDP, exhibited Fab-Fab distances of both 3.8 nm and 5.25 nm, and was used to construct a potential model of the aligned Fab domains forming a lattice perpendicular to the lipid bilayer (Fig. 1d, f). This model provided the potential interactions between adjacent heavy chains, but within the array there will also be interacting light chains required to extend the array in two dimensions. Although this interaction is not captured within 5TDP, it is present in other crystal structures (Supplementary Fig. 6). The presence of all the necessary interacting partners in multiple deposited structures supports our finding that these can synergise on lipid membranes to provide the interactions required for formation of an extended 2D Fab array, as observed here.

### Elevated hexameric IgG3 platforms activate the human complement cascade
IgG1-Fc and IgM-Fc domains form hexameric platforms 11 nm and 13.5 nm above liposome surfaces, respectively[24,30] (Fig. 1a). However, in tomograms of IgG3 bound to antigenic liposomes, elevated platforms $22 \pm 3.5$ nm (SD, $n = 100$) above and parallel to the liposome surface were visible above the Fab domains (Fig. 1b and Supplementary Fig. 4b). IgG3-Fc platforms are therefore an additional 11 nm further from the antigenic membrane than for IgG1, which is possible due to the long hinge of IgG3 adopting an extended conformation (Supplementary Fig. 1a and Supplementary Table 2). These platforms were often observed above ordered, striated IgG3-Fab arrays; from a total of 1193 elevated platforms, 525 were above ordered, striated Fab domains, with the remainder above disordered density (Fig. 1b and Supplementary Fig. 4b).

To determine whether these platforms represented hexameric IgG3-Fc domains, we performed subtomogram averaging and 3D classification of all 1193 particles, which yielded a hexagonal Fc platform (Fig. 2a and Supplementary Fig. 7). This Fc geometry is in keeping with previous data of C1-bound IgG1 platforms[22,30], although these

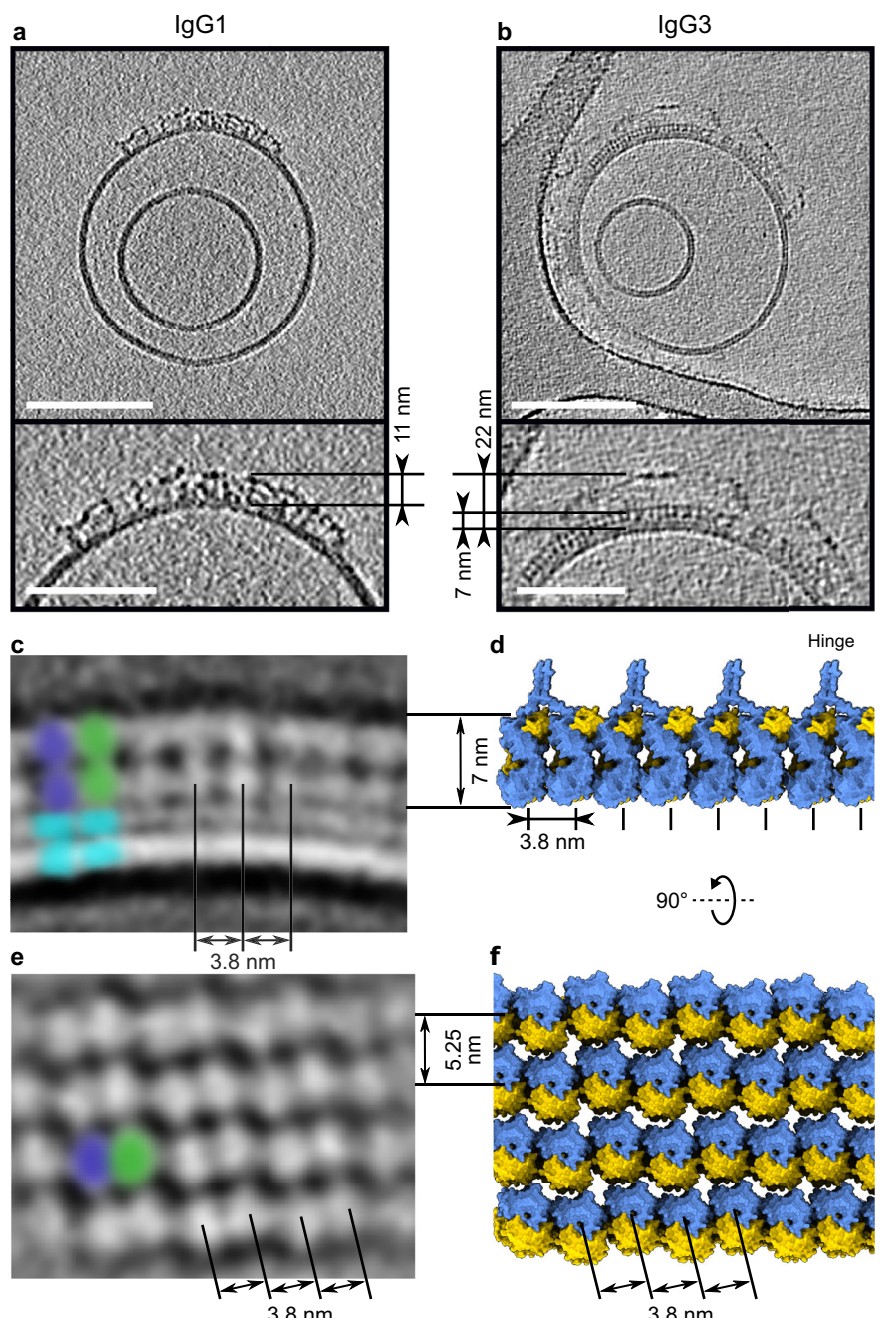

**Fig. 1 | Characterisation of antigen-bound IgG1 and IgG3. a, b** Slices 10 nm thick through tomograms of antigen-bound IgG1 (**a**) and IgG3 (**b**). Magnified regions (bottom) with membrane-to-Fc platform measurements. Scale bars represent 100 nm (top) and 50 nm (bottom). **c** Slice 5 nm thick through a subtomogram average of the ordered Fab domains viewed from the side. Two Fab domains are coloured violet and green, and membrane denoted as cyan. Repeating distance of 3.8 nm and 7 nm height of striations are shown. **d** Possible model of the Fab array from the same orientation as (**b** and **c**). Blue and yellow represent the heavy chain and the light chain of IgG3 Fab domains, respectively. **e** Slice 5 nm thick through a subtomogram average of aligned Fab domains observed at the apex of liposomes (Supplementary Fig. 5c, d). Two Fab domains are coloured violet and green, and repeating distances of 3.8 nm and 5.25 nm shown. **f** Possible model of the Fab array from the same orientation as (**e**), effectively rotated 90° from (**b** and **c**). Heavy and light chains are blue and yellow, respectively.

hexagonal Fc platforms have not been previously observed for non-mutated or antigen-bound antibodies in the absence of the C1 complex. Our data did not resolve the hinge regions and so, to yield a complete model of the IgG3 hexamers, we modelled the hinge as a semi-rigid helix which spanned the ~11 nm between the Fc-hexamer and Fab domains. The Fc platform was rotated to minimise the average hinge length between the respective Fc and Fab domains, yielding the complete hexameric IgG3 structure in Fig. 2b. The elevated platforms and divalent binding are therefore both enabled by the flexibility of IgG3-hinge region compared to IgG1 (Supplementary Table 2). Next,

NHS was added as a source of complement proteins to either IgG1- or IgG3-coated antigenic liposomes and maintained at 4 °C before vitrification to stall the classical complement pathway at C4b deposition[24,31]. Tomogram slices showed additional density on top of the Fc platforms of both IgG1- and IgG3-opsonized liposomes that was consistent with the C1 complex (Fig. 3a, b and Supplementary Fig. 8). In the case of IgG3, C1 was only observed on elevated Fc platforms, which were an average of 19 ± 3 nm (SD, $n = 100$) from the membrane, similar to without C1. Incubation of liposomes with IgG and NHS at ambient temperatures before vitrification revealed extensive opsonization of

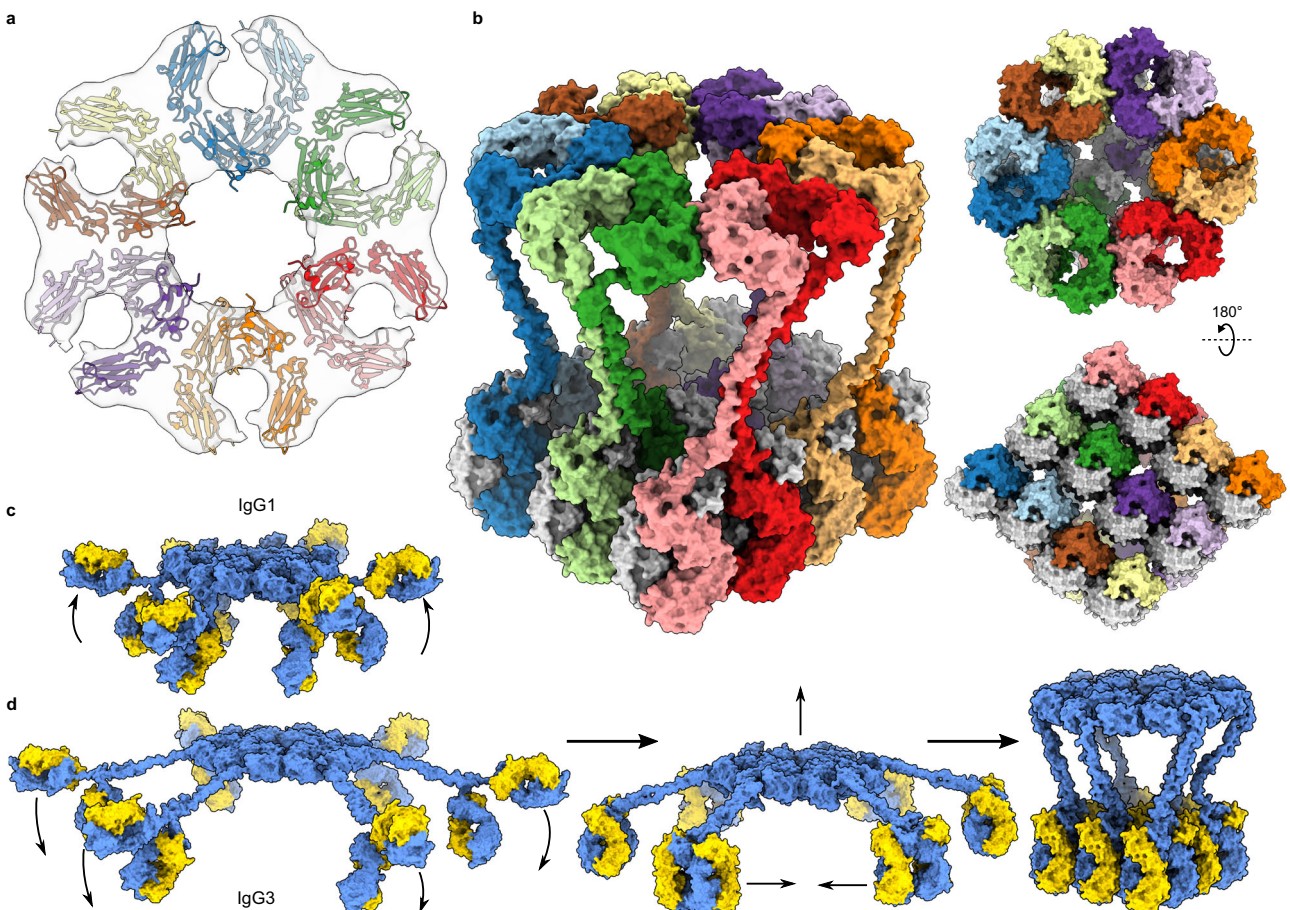

**Fig. 2 | A model of antigen-bound hexameric IgG3. a** Hexameric IgG3-Fc platform C6-symmetric map (grey) and model (coloured ribbons). **b** Complete model of IgG3 hexamer viewed from three different directions. Light chains are grey, each heavy chain pair is a separate colour. **c** Proposed IgG1 hexamer showing monovalent Fab binding. The second Fab is oriented away from the surface (arrows).

**d** Proposed IgG3 hexameric models; the longer upper hinge region allows either monovalent (left) or divalent (middle) antigen binding. Divalent Fabs associate (middle) and form an array, causing the Fc platform to become elevated (right). For (**c** and **d**), the heavy chain is blue and the light chain is yellow.

the membranes with complement proteins and formation of MAC pores, which together form a protein corona (Fig. 3c, d and Supplementary Fig. 9). In the case of IgG3, Fab arrays and elevated Fc platforms were both visible, and C1 was observed again only on elevated platforms. For IgG1, the 11 nm-high platforms were found within the opsonizing coat (Fig. 3c and Supplementary Fig. 9c), whereas the elevated IgG3-Fc platforms were generally above the protein corona (thickness varies from ~10 to 25 nm; Fig. 3d and Supplementary Fig. 9d).

Subtomogram averaging of all IgG3-C1 complexes for samples incubated at 4 °C before vitrification yielded a low-resolution map (Supplementary Fig. 10), which revealed the presence of a hexameric IgG3-Fc platform, all components of the C1 complex (C1qr$_2$s$_2$), as well as C4b, a product of C1s (Fig. 3e, f). Regions of lower resolution were likely caused by structural heterogeneity and/or flexibility; this was especially apparent for the Fab domains, which remained unresolved. To address this, we performed focused refinements around the C1 complex, IgG3-Fc-gC1q region, C1r$_2$s$_2$ platform, and C1s-C4b-Fab region (Supplementary Fig. 10b).

### C1r-s interactions provide insights into C1 autoactivation
Focused refinement of the masked IgG3-Fc-C1 region reached 28 Å resolution, which was sufficient to build a model of the Fc platform and C1 complex (Fig. 3e, f, Supplementary Fig. 10 and Supplementary Fig. 11a). C1q adopted a similar structure as found in complexes of C1 bound to IgG1 and IgM, with all 6 globular C1q head domains (gC1q)

bound to a hexameric IgG3-Fc platform and separated by ~9 nm on each side of the hexagonal Fc platform (Supplementary Fig. 11b)[22,24]. In contrast to IgG1 and IgM, the C1q stalk was more upright on IgG3 (Supplementary Fig. 11c). The C1r$_2$s$_2$ protease platform, composed of 'complement C1r/C1s, Uegf, Bmp1' (CUB)-1, epidermal growth factor (EGF) and CUB2 domains (Supplementary Fig. 1b), was located between the C1q collagen arms (Fig. 3g and Supplementary Fig. 11d), as found in other models of C1[22,24].

Both C1r and C1s arms, comprising complement control protein (CCP)-1/2 domains and the SP domain, were resolved (Fig. 3g, h and Supplementary Fig. 11d). The two C1r arms were oriented parallel to the liposome membrane (Fig. 3e, f and Supplementary Fig. 11), and one of the C1s arms was oriented down towards the IgG3-Fc platform where it interacted with C4b, similar to observations of antigen-bound IgM-C1-C4b complexes[24]. However, there was reduced density for the opposing C1s arm in this configuration, which was also adjacent to additional density not accommodated by the model (black arrow, Fig. 3g). This additional density also corresponds to the C1s arm (Fig. 3g, h) but, although further focused refinement improved the resolution to 25 Å, classification was unable to separate the two populations (Supplementary Fig. 10). The map therefore represents a superposition of C1s configurations, with one state representing C1s extending towards the IgG3-Fc platform, and another state showing the C1s arm adopting a configuration found closer to the C1r SP domain (4 nm *cf.* 16 nm; Fig. 3h), potentially adopting the configuration of the C1 proteases during complement initiation. Empty density

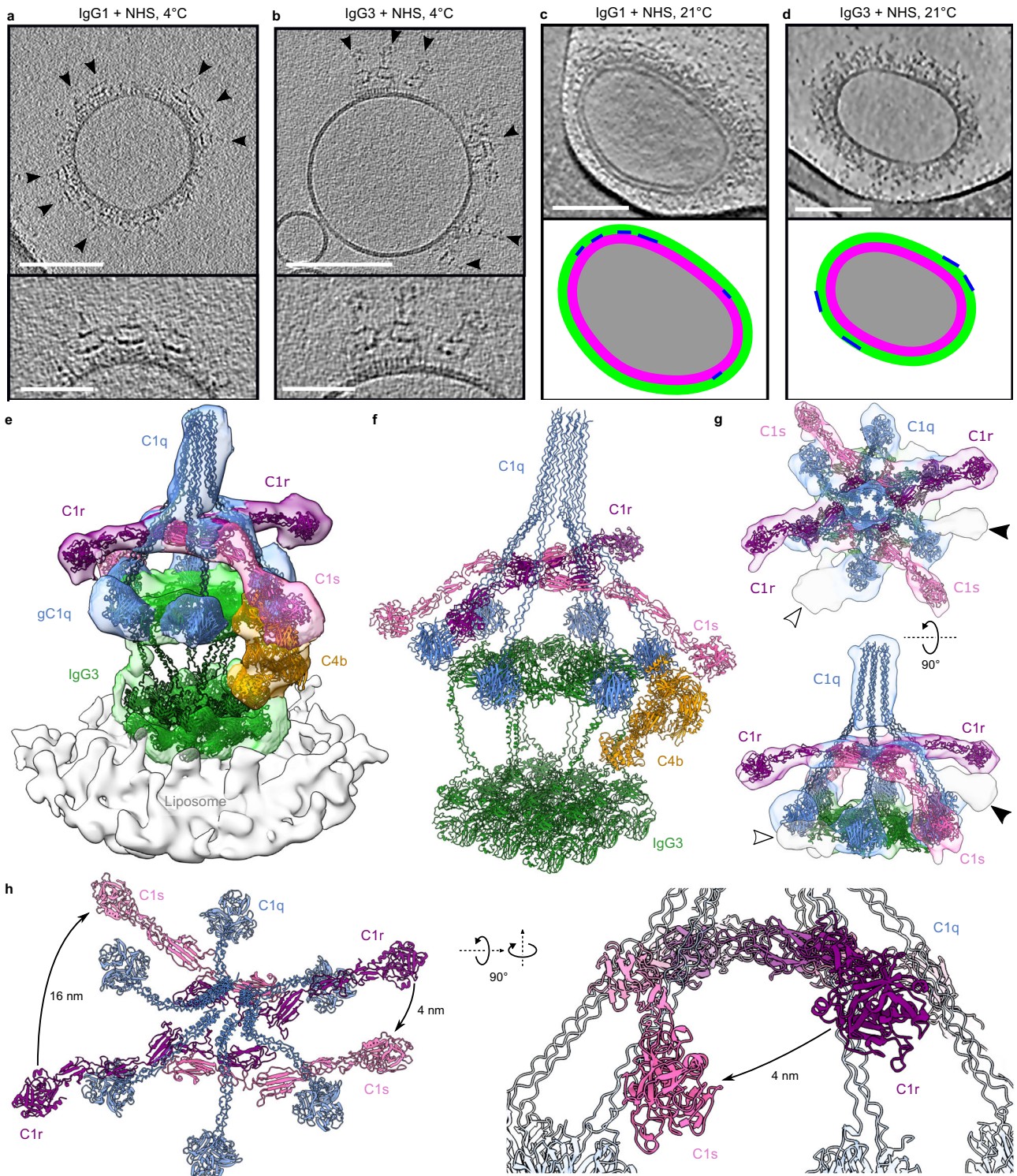

**Fig. 3 | Structures of IgG3-C1-C4b on antigenic liposomes. a, b** C1 complexes (black arrowheads) on IgG1-coated liposomes (**a**) and IgG3-coated liposomes (**b**). Scale bars represent 100 nm (top) and 50 nm (bottom). **c, d** Denoised tomographic slices 10 nm thick of opsonized liposomes coated with IgG1 (**c**) and IgG3 (**d**) (top). Schematic diagrams (bottom) show the location of antibody platforms (blue lines) on the liposomes (grey). Pink and green represent 12 nm and 24 nm-thick heights of the protein corona. Scale bars represent 100 nm. **e, f** Subtomogram map (left) and fitted model (right) of the complete IgG3-C1-C4b complex. C1q, blue; C1r, purple; C1s, pink; C4b, orange; IgG3, green; liposome, grey. **g** Focussed refinement of the C1 complex showing empty density near the Fc domain (white arrow) and adjacent to one of the C1r SP domains (black arrowhead). **h** Model built into the sub-tomogram map showing the C1s arm adopting a conformation close to the C1r SP domain, annotated with measured distances.

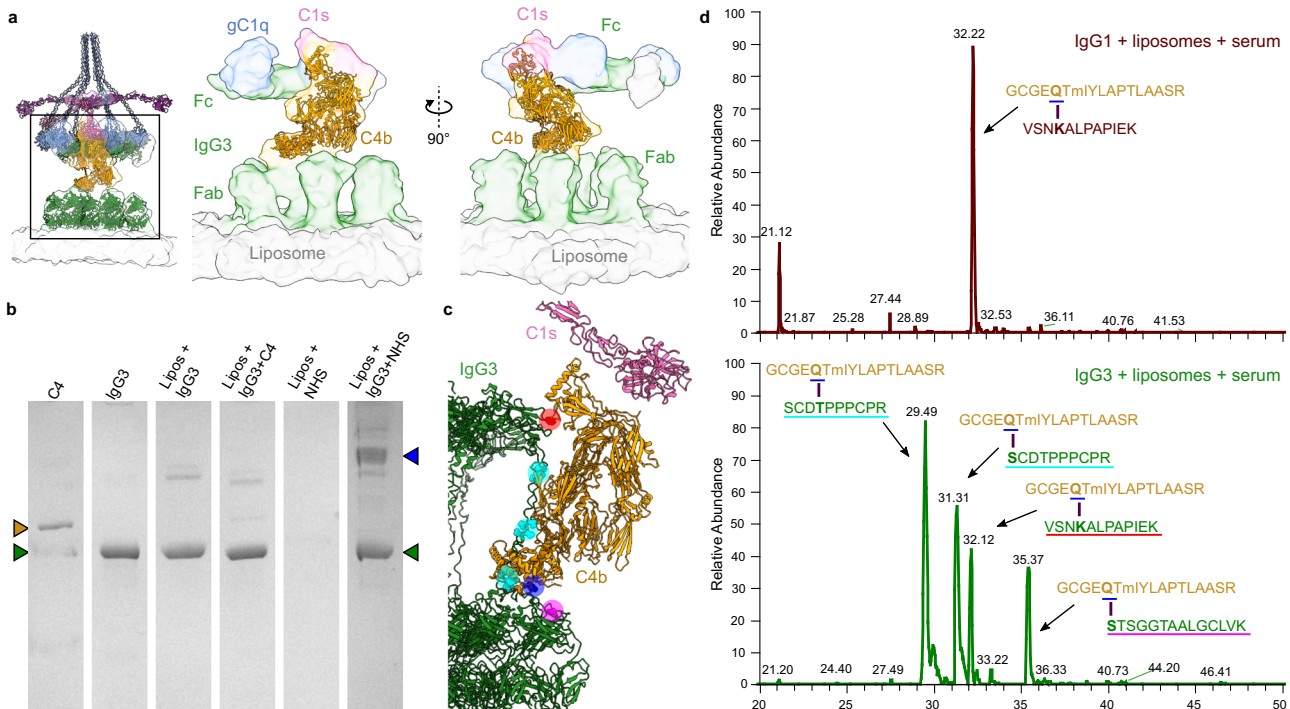

**Fig. 4 | CryoEM and MS characterisation of C4b binding to IgG3. a** Density corresponding to C4b (orange) reveals a tilted molecule bound to IgG3 Fab domains (green). gC1q are displayed in blue, C1s is displayed in pink and the liposome surface is displayed in grey. **b** Example of a Coomassie-stained gel showing purified IgG3 (green arrow), C4b (orange arrow), and mixtures with liposomes (Lipos) and human serum (NHS). A new band appeared upon mixture of all required components (blue arrow). Gel was repeated three times. **c** Model of C4b bound to IgG3 showing sites for covalent attachment as determined using MS. Sites of attachment are indicated with the following colours: **S**TSGGTAALGCLVK, pink;

**S**CDTPPPCPR, cyan; VSN**K**ALPAPIEK, red. Residues identified by MS that attach to C4b are denoted as spheres. The C4b-TED reactive glutamine residue is shown as a blue sphere. **d** MS analysis of tryptic peptides of C4b (GCGE**Q**TmIYLAPTLAASR, orange; m, oxidised methionine) crosslinked to IgG1 (upper, red) and IgG3 (lower, green). For both samples, extracted ion chromatograms show traces for the C4b peptide (orange) attached to VSN**K**ALPAPIEK (32 min). Only IgG3 has C4b conjugated to other sites. Crosslinked residues are denoted in bold and underlined in the colour corresponding to (**c**).

## C4b is deposited onto IgG3 at distinct sites

Classification of the Fab region did not reveal any single structure, and extensive 3D classification was unable to resolve distinct configurations of IgG3-Fabs or hinge regions. Instead, the model constructed above of IgG3 alone (Fig. 2b) was placed in the density (Fig. 3e, f). Focused refinement around the IgG3-Fc region revealed the presence of additional density between C1s and the Fab layer (Fig. 3e, f, Supplementary Figs. 10b and 11e). Although relatively low (~30 Å) resolution, this density clearly corresponded to C4b, and further focused refinement yielded a map containing part of the IgG3-Fc platform, densities corresponding to Fab domains, and C4b attached to part of C1s (Fig. 4a, Supplementary Figs. 10 and 11e). C4b was previously observed in complexes IgM-C1-C4b, where the upper C4b C-terminal complement (CTC) domain was bound to the C1s protease and the thioester domain (TED) attached to the liposome surface[24]. The TED contains the reactive thioester of C4 and is responsible for covalent attachment to membranes and proteins after cleavage to C4b[23,24,32]. However, in this map, the C4b TED was ~8 nm away from the liposome surface, and instead appeared to be bound to the upper IgG3-Fab or hinge regions (Fig. 4a, c and Supplementary Fig. 11e). Furthermore, C4b was tilted ~22° away from vertical (Supplementary Fig. 11e), whereas C4b was vertical in IgM-C1-C4b complexes[24].

The resolution of our subtomogram maps was not sufficient to identify the location of C4b covalent attachment to IgG3. To explore this interaction, antigenic liposomes were opsonized with IgG1 or IgG3

was also located adjacent to the IgG3-Fc hexameric ring (white arrow, Fig. 3g), which may be due to neighbouring IgG3 complexes, as previously observed for IgG1[22].

before NHS was added and incubated at 4 °C to halt the cascade at C4b deposition[24,31]. Liposomes, together with any bound proteins, were then purified at 4 °C and proteins visualised on a Coomassie-stained SDS-polyacrylamide gel under non-reducing conditions. In the presence of all components necessary for C4b deposition, a band of high molecular weight appeared (Fig. 4b and Supplementary Fig. 12a). Bands were excised, digested using in-gel trypsin treatment, and the resulting peptides analysed using mass spectrometry (MS). Multiple cross-linked peptides between the tryptic peptide GCGE**Q**TMIY-LAPTLAASR from C4b (where **Q** is the glutamine residue that mediates covalent attachment) and serine, threonine and lysine residues in tryptic peptides from IgG were identified. All figures show data from samples with oxidised methionine (L20222001054a, Supplementary Data 1), although samples with non-oxidised methionines were analysed and identical results were observed (Supplementary Data 1). In the case of IgG3, the C4 peptide was primarily found linked to the peptides SCDTPPPCPR (which forms a triple repeat within the IgG3 hinge), STSGGTAALGCLVK (part of the IgG-Fab fragment, in structural close proximity to the hinge region), and VSNKALPAPIEK (part of the IgG-Fc region) (Fig. 4c, d and Supplementary Figs. 12–14). In addition, a small fraction (based on the signal in the raw data) was linked to the peptide CPEKSCDTPPPCPR (Supplementary Data 1), representing a longer variant of the peptide derived from the IgG3 hinge (Supplementary Tables 1 and 2).

The IgG3 hinge comprises a triple repeat with identical sequences, and as such the exact position of C4b covalent attachment cannot be determined by MS. However, our subtomogram map shows likely attachment to the first instance, closest to the Fab domain (Fig. 4a, c). Within this triple repeating hinge sequence, three peptides could be

identified by MS, with either the serine (peak at 31.31 min) or the threonine (peak at 29.49 min) as the site of C4b covalent attachment to **S**CD**T**PPPCPR (Fig. 4d and Supplementary Fig. 12b), or the lysine residue in the less abundant CPE**K**SCDTPPPCPR peptide (peak at 29.63 min, Supplementary Data 1). These same threonine residues in the triple-repeat IgG3 hinge region sequences are also the location of the O-linked glycans which occupy ~10% of the six available sites (Supplementary Fig. 1a)[8], and were also detected in our IgG3 (Supplementary Fig. 15). Although no O-linked glycans were detected on the tryptic peptides linked to C4b, we cannot discount the possibility that the glycans are also sites for C4b deposition. C4b was also detected bound to the IgG3-Fab domain, to the first serine within the peptide **S**TSGGTAALGCLVK (Fig. 4d and Supplementary Fig. 13). In our model, this peptide is located at the interface of the IgG3 Fab and hinge, and within the region of C4b attachment suggested by our map (Fig. 4c).

One of the sites of C4b deposition onto IgG identified by MS was not proximal to the TED in our model. The peptide sequence VSN**K**ALPAPIEK is located at the periphery of the Fc-$C_H2$ domain, adjacent to the location of C1q binding, with C4b attached to the first lysine residue (Fig. 4c, d and Supplementary Fig. 14). Interestingly, this was also the only location that C4b was detected attaching to IgG1 (Fig. 4d and Supplementary Fig. 14). Incubation of liposomes, IgG3 and purified C4 revealed no direct interaction of C4 with antibodies (Fig. 4b, Supplementary Figs. 12a and 16), indicating that there is no spontaneous cleavage of C4 in the absence of C1, and that the Fc domains are target sites for C4b deposition only after cleavage by the C1 complex. In IgG1, this lysine is labelled as K326, and is critical for C1 binding and complement activation[33]. These data suggest a mechanism for C4b deposition, discussed below.

## Discussion

Imaging antigen-bound IgG3 on fluid liposome surfaces revealed a propensity for Fab domain clustering and formation of an ordered array (Fig. 1b and Supplementary Fig. 4b). Although IgG1 and IgG3 differ predominantly in the hinge region (Supplementary Fig. 1a, Supplementary Tables 1 and 2), ordered Fab domains were only observed in tomograms of antigen-bound IgG3. IgG1 and IgG3 Fab domains contain only 3 differences; S131C, K133R, and I199T (Supplementary Fig. 5e and Supplementary Table 1) (Eu numbering[34]; residue numbering is equivalent between IgG1 and IgG3 before the hinge region). Of these, only I199T is located on the surface of the Fab domain, but is not well positioned to impact Fab-Fab binding. It is therefore more likely that the different hinge regions explain the ability of IgG3-Fabs to form ordered arrays. All IgG subclasses comprise two Fab domains and are capable of divalent antigen-binding. However, IgG1 forms hexameric Fc platforms when undergoing monovalent binding to surface antigens. Furthermore, enforcing monovalent binding by forming bispecific IgG1 was found to improve complement activation of other antigen-antibody complexes[30]. The short upper hinge region of IgG1, between the Fab and the first interchain disulfide bridge, comprises only 5 residues (Supplementary Table 2), which introduces structural constraints that limits simultaneous divalent binding and Fc oligomerization[6], such that coordinated divalent Fab binding is inhibited in IgG1 hexamers (Fig. 2c). Indeed, Fab domains are separated by ~120° in structures of monomeric IgG1[6,35]. In contrast, the upper hinge region of IgG3 is 9 residues long (Supplementary Table 2). These additional 4 residues impart significant flexibility and potential separation between the IgG3-Fab domains[6,36,37], which consequently allows divalent Fab binding and concurrent formation of an ordered Fab array along the lipid membrane, even in the presence of hexameric Fc platforms (Fig. 1 and Supplementary Fig. 4b). This divalent binding of IgG3 may enhance the binding avidity of hexamers by increasing the maximum number of antigen-bound Fab domains from 6 to 12 compared to IgG1 hexamers. The ordered Fab arrays observed here also allow IgG3 monomers to pack closer together on a membrane than IgG1, increasing the density of antibodies and potentially enhancing downstream effector functions (Fig. 5a). Indeed, IgG3 is associated with improved neutralisation compared to other IgG subclasses;[12] this was attributed to the F(ab′)$_2$ fragment, which includes the upper hinge of IgG3, and the difference between IgG3 and IgG1 was lost when using monomeric Fab fragments[38], in agreement with our structural data.

We observed that IgG3 formed Fc platforms ~22 nm ± 3.5 nm (SD, $n = 100$) above the liposome surface, with the hinge adopting an extended conformation (Fig. 1b and Supplementary Fig. 4b). Previous models of IgG3 binding and complement activation have implied a wide, low binding mode, where the extended IgG3 hinge allows Fab binding to widely-spaced antigens (Fig. 2d), which has been used to explain the more efficient complement-activating ability of IgG3 on low antigen-density or rigid surfaces[39]. Binding to widely-spaced antigens would cause a concomitant reduction in the height of the hexamerised Fc platform (Fig. 2d). Whilst our model does not preclude this, our data reveals a propensity for IgG3-Fab domains to cluster close together that, combined with the semi-rigid disulfide-linked core hinge[1,40], causes the Fc platform to rise ~22 nm up away from the

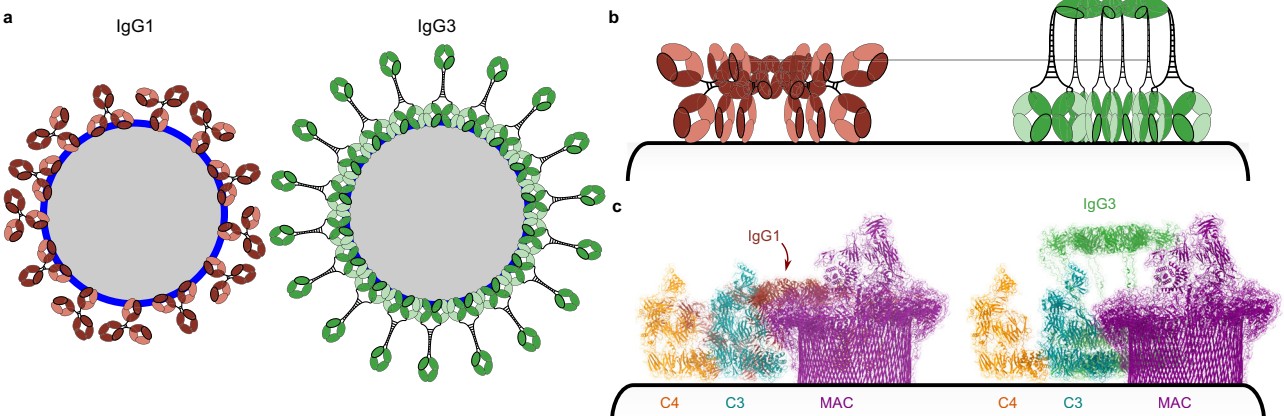

**Fig. 5 | Functional consequences of ordered IgG3-Fab and elevated IgG3-Fc domains. a** Neutralisation by IgG1 (left, red) and IgG3 (right, green). IgG3 can pack much closer together than IgG1 by forming a Fab array and cover more of the pathogen surface (blue), potentially enhancing pathogen neutralisation. **b** The higher IgG3 (green) platform is more exposed than IgG1 (red) on opsonised membranes, allowing greater FcγR access. **c** Height comparisons of various complement components with IgG3 and IgG1 on a surface. C4, orange (PDB code 4XAM); C3, cyan (PDB code 2I07); IgG1, red; IgG3, green; MAC, purple (PDB code 6H03).

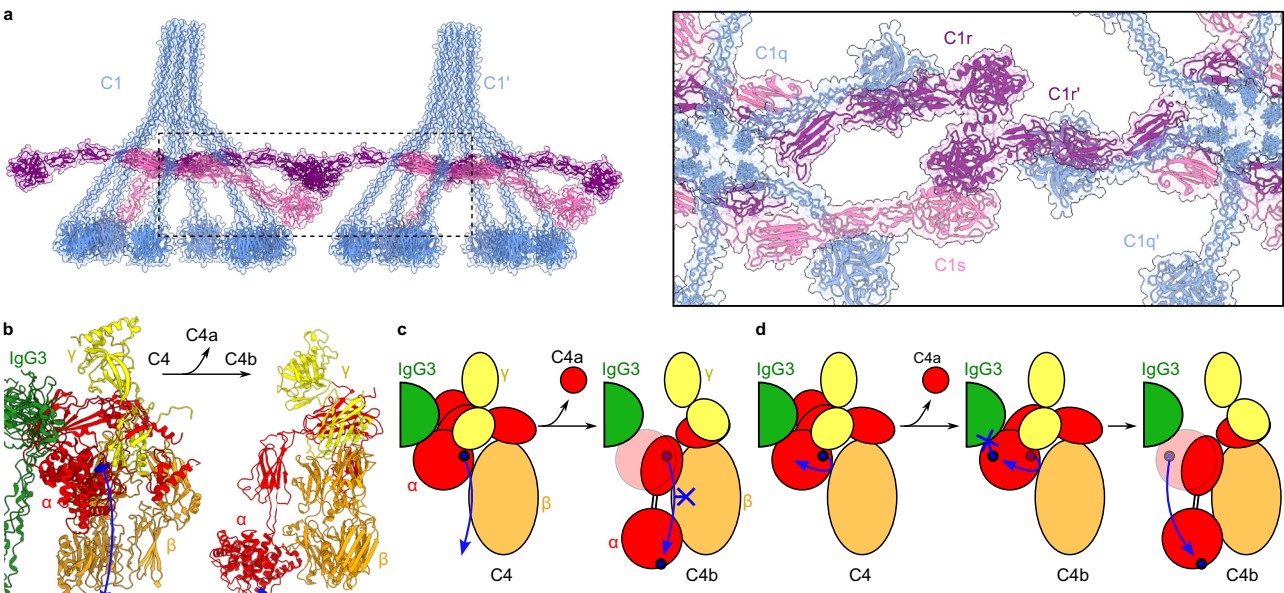

**Fig. 6 | Insights into C1s activation and C4b deposition by IgG3-C1 complexes. a** Cross activation by C1 complexes. The C1r arm (blue) of one complex can react with the C1r' arm (blue) of the adjacent complex. The conformation of C1s (pink) on IgG3 platforms also enables activation by the C1r' protease (purple) of an adjacent complex. **b** Crystal structures showing the large, 9 nm, rearrangement of domains after cleavage of C4 (PDB code 5JPM) to form C4b (PDB code 4XAM). Domains are coloured as follows: α-chain, red; β chain, orange; γ chain, yellow. **c** The direct route

from C4 to C4b (blue arrow) would orient the reactive thioester (blue dot) towards the β chain, potentially enhancing intramolecular conjugation (blue cross), and formation of an inactive C4b molecule. **d** Instead, if an indirect route is taken, the thioester would first be oriented away from the rest of C4b (blue arrow), before formation of the final C4b structure. This would also orient the thioester towards IgG3-Fc, resulting in crosslinking to lysine K326 (blue cross), as seen by MS.

antigens (Fig. 1b and Supplementary Fig. 4b). On crowded cell membranes, these elevated IgG3-Fc domains will be much more exposed than for shorter IgG1 (Fig. 5b, c). In this study, elevated IgG3 platforms were observed on liposomes coated with complement proteins (Fig. 3d and Supplementary Fig. 9), whilst the shorter IgG1 platforms were apparently obscured by the surrounding corona of complement proteins (Fig. 3c and Supplementary Fig. 9). Modelling this arrangement reveals how IgG1 hexamers form Fc platforms that are found below the height of opsonizing complement components (red arrow, Fig. 5c), whilst IgG3 is above these proteins, and therefore more exposed for binding. This greater exposure may enhance the ability of IgG3 to bind to receptors, such as Fcγ-receptors (FcγRs), on heavily opsonized surfaces or when the antigen is close to the membrane and deep within the protein layer that covers cells. This may be a functional explanation for the observation that IgG3 binds some FcγRs more strongly than IgG1 in certain environments[41,42].

C1 was observed adopting a similar configuration to IgG1-C1 and IgM-C1 complexes[22,24], with C1q binding to a hexagonal platform with 6 binding sites spaced ~9 nm apart (Supplementary Fig. 11b), although with a more upright stalk (Supplementary Fig. 11c). The apparent diameter of the IgG3-Fc hexameric platform was unaffected by C1 binding. Due to the increased flexibility of IgG3 compared to IgG1 and IgM, the EM map reached lower resolution. However, it was still possible to identify two configurations of the C1s arms; one reached down to C4b, whilst the opposing arm was oriented toward the adjacent C1r SP (Fig. 3g, h). This flexibility is possible for C1s due to the absence of collagen-binding between C1s-CUB2 and C1q at this location[43], which is in contrast to the C1q collagen-binding sites found C1r-CUB1 and CUB2, and C1s-CUB1, which are observed to be less structurally heterogeneous. This C1s flexibility is reflected in the variable C1s configurations previously observed[24,44,45]. The C1 complex is activated by auto- or cross-catalytic cleavage when C1r cleaves both C1r and C1s, which then propagates the classical complement cascade. A similar configuration of C1s adjacent to C1r was previously posited based on density of the CUB2

domain in IgG1-C1 complexes[22], the orientation of which suggested that C1r may adopt a configuration close to C1s. Here, the C1s arm changes configuration and is located only 4 nm from the C1r SP when not interacting with C4b (Fig. 3h), potentially representing the conformation that C1s adopts while undergoing cleavage of the scissile bond and protease activation by C1r within the same C1 complex. Furthermore, the more upright C1q stalk observed here, compared to structures of IgG1 and IgM (Supplementary Fig. 11c), may indicate that the stalk tilts less during C1r/s activation than after C1s cleavage or during C4b deposition. (Fig. 6a). Such cross-activation was implied from cryoEM maps of IgG1-C1 clusters on liposome surfaces[22], and cross activation is required by the homologous mannan-binding lectin and associated serine protease (MBL-MASP) complexes[46]. Indeed, weak density potentially from neighbouring IgG3 complexes was present in our maps (white arrow, Fig. 3g), indicating cross-activation of IgG3-C1 complexes is sterically possible. Cross activation would rely on C1 complexes binding close to one another or, as in this case, to antibody-antigen complexes that are able to diffuse laterally in the fluid lipid bilayer.

Membrane fluidity and antigen diffusion was proposed to allow antibody rearrangement to allow C4b deposition from IgM-C1 complexes[24]. Whether such a mechanism is also necessary or applicable here could not be determined from structural data. However, the extended hinge of IgG3 caused deposition of C4b by C1s onto the antibody itself (Fig. 4a and Supplementary Fig. 11e). Previous studies have also described C4b binding to IgG aggregates, although the exact binding site of C4b to these IgG aggregates could not be localised[47,48]. However, by using cryoEM C4b was previously observed being deposited onto lipid membranes by IgM-C1 complexes[24], but has not been seen in complexes of IgG1-C1[22]. Although complement progression indicates that C4b is also deposited on to lipid membranes, allowing formation of C3 convertases (Supplementary Fig. 2c, d), the height of IgG3-C1 meant that the lipid bilayer was further from C1s than the length of C4b (~18 nm and ~14 nm, respectively), and consequently C4b was deposited onto one of three sites on IgG3 as well. Two of these

sites were adjacent and located at the top of the Fab domain/upper hinge interface (Fig. 4c, d). This location is a similar distance from C1s as the lipid membrane in IgM-C1 complexes. Remarkably, C4b was not deposited randomly at this location, but instead at one of 4 distinct sites; S134, K235 (although to a lower extent than for the others), S236, and T239 (Fig. 4c, d, Supplementary Table 1 and Supplementary Data 1). These residues all contain sidechains with either a hydroxyl or amine group, which are available for C4b conjugation[32]. One further site was identified, distal to the region identified by cryoEM, which was on the IgG-Fc domain and occurred for both IgG1 and IgG3. Here, C4b was deposited onto K326 (Fig. 4d and Supplementary Table 1), presumably via the lysine sidechain amine. Our cryoEM model of hexameric IgG3 Fc platforms revealed the location of K326 on the periphery of the Fc domain, within the $C_H2$ dimers (Fig. 4c). In the complete hexameric-IgG3-C1-C4b, this site is far from the reactive thioester-containing TED of C4b[23] (Fig. 6b, c), and it is unclear how the Fc domain is bound. One explanation is an intermediate step after C4 cleavage by C1s that first rotates the TED away from the macroglobulin domains of C4b, before the TED descends ~9 nm to the final position found in structures of C4b (Fig. 6d)[23]. This would have the benefit of keeping the reactive thioester away from the rest of the C4b molecule during this structural rearrangement, thereby preventing covalent attachment to itself (Fig. 6c, d). This would also cause orientation of the TED towards the Fc periphery, resulting in attachment to the proximal lysine amine group as observed here.

C4b was deposited directly onto lipid bilayers in IgM-C1 complexes in a vertical orientation[24]. In contrast, in complexes of IgG3-C1, the C4b molecule was tilted ~22° away from vertical (Fig. 4a, c and Supplementary Fig. 11e). This is presumably due to the height of the IgG3-Fc platform, which can only accommodate C4b if it is not completely vertical. How this C4b location affects complement regulation remains to be determined. Membrane-bound complement regulators, such as membrane cofactor protein (MCP)[25], may not be as effective when C4b is not colocalised on the membrane. In contrast, soluble regulators, such as C4b-binding protein (C4BP) and Factor I, or extended molecules, such as complement receptor 1 (CR1)[25], may have more access to IgG3-associated C4b. Furthermore, hinge length is not linearly correlated with complement activation; IgG3 hinges can be too short or too long[14]. This may be due to C4b being deposited either on the lipid bilayer (for short hinges found in IgG1), or the antibody Fab/hinge region (for long hinges found in IgG3). Hinge lengths shorter than IgG1, longer than IgG3, or between these two values, may have impaired C4b deposition and therefore compromised complement activation. This could also depend on the location and environment of the antigen, and there is evidence that different antigens may have different ideal hinge lengths for C4b deposition[17,49], which can now be explained by our structural data. It will be intriguing determine whether there is a correlation between antigen size and epitope location with complement activation by antigen subclass or IgG3 allotype.

The more potent effector functions of IgG3 are generally attributed to the longer hinge of IgG3 compared to other antibodies[12,14,16]. Human IgG3 allotypes represent enormous variability in the length of the hinge, and more than 15 allelic variations are currently known[9,50], with hinge lengths varying from 32-62 amino acids. Furthermore, O-linked glycosylation at three sites, each of which are partially occupied, may also contribute to IgG3 hinge rigidity and hence persistence length[8]. How these allotypes and glycoforms contribute to IgG3 effector functions and C4b deposition remains unclear, although our data provide structural insights into the role of the IgG3 extended hinge.

## Methods
### Production and purification of antibodies
Recombinant anti-DNP IgG antibodies are based on variable domains mouse monoclonal antibody G2a2 against the hapten dinitrophenol (DNP)[51], combined with human constant domains. Isotype variants were based on human IgG1 (G1m(f) isotype) and IgG3 (P01860), all variants were co-expressed with kappa light chain. Gene constructs for heavy and light chains were designed to include a cleavable signal sequence for secretion and were flanked by restriction sites to allow for simple DNA cloning. DNA fragments for variable and constant domains of heavy and light chains were ordered separately from GeneArt (Thermo Fisher Scientific, Waltham, Massachusetts, USA) and cloned into a pcDNA3.3 vector (Thermo Fisher Scientific, Waltham, Massachusetts, USA) behind a CMV promotor.

Antibodies were produced by transfecting 12.5 μg each of heavy chain plasmid and light chain plasmid in Expi293F™ (catalogue number A1427; Thermo Fisher Scientific, Waltham, Massachusetts, USA) cells using the ExpiFectamine™ 293 Transfection Kit (Thermo Fisher Scientific, Waltham, Massachusetts, USA) following the manufacturer's instructions. After 5–7 days, transfection broth was centrifuged, the supernatant harvested and sequentially filtered through 0.45 μm and 0.2 μm filters (Cytiva, Marlborough, Massachusetts, USA). Antibodies were then purified on protein A (Genscript, Piscataway, New Jersey, USA) or in the case of IgG3 on either a protein A/G (Thermo Scientific Scientific, Waltham, Massachusetts, USA) or protein L cartridge (Cytiva, Marlborough, Massachusetts, USA) followed by concentrating and re-buffering to PBS using 30 kDa Amicon centrifugal filters (Merck Millipore Sigma, Burlington, Massachusetts, USA). Concentration of the purified antibody preparations was determined by NanoDrop™ 2000c spectrophotometer (Thermo Fisher Scientific, Waltham, Massachusetts, USA).

Antibody subclasses, at 0.350 mg/ml in PBS, were further purified and characterised using a Superdex™ 200 Increase 3.2/300 column (Cytiva, Marlborough, Massachusetts, USA), which was equilibrated with PBS on an Äkta™ pure system (Cytiva, Marlborough, Massachusetts, USA) (Supplementary Fig. 2a). Purified samples were further analysed using polyacrylamide gel electrophoresis (PAGE). For non-reduced conditions, samples were directly mixed with Laemmli buffer (Bio-Rad Laboratories B.V., California, USA) and loaded onto a 4–12% pre-cast Bis-Tris Protein Gels (Bolt™, Invitrogen™, Thermo Fischer scientific, Waltham, Massachusetts, USA). For reduced conditions, samples were mixed with dithiothreitol (0.1 M final concentration; DTT; Thermo Fisher Scientific, Waltham, Massachusetts, USA) and Laemmli buffer and heated to 99 °C for 10 min before loading. Gel electrophoresis was performed in MOPS buffer (pH 7.7, Sigma-Aldrich, St Louis, MO, USA) supplemented with 0.1% w/v sodium dodecyl sulfate (SDS; Merck, Massachusetts, USA) on a Invitrogen™ Mini Gel Tank electrophoresis device (Thermo Fisher Scientific, Waltham, Massachusetts, USA) for 35 min and 75 min for the reducing and non-reducing conditions, respectively. Gels were stained using SimplyBlue™ SafeStain (Invitrogen, Thermo Fisher Scientific, Waltham, Massachusetts, USA) for 20 min and de-stained in deionised water overnight before imaging using a white light conversion screen in a ChemiDoc XRS+ (Bio-Rad Laboratories B.V., California, USA) (Supplementary Fig. 2b).

### Liposome production
All lipids were purchased from Avanti Polar Lipids (Alabama, USA). Liposomes displaying 1 mol% DNP and encapsulating 20 mM sulforhodamine B (Sigma-Aldrich, St Louis, MO, USA), composed of dimyristoylphosphatidylcholine (DMPC), dimyristoylphosphatidylglycerol (DMPG), cholesterol and dinitrophenyl-cap-dipalmitoylphosphatidy lethanolamine (DNP-cap-PE) (44:5:50:1 mol%) in PBS were generated as previously described[27].

### Liposome-based complement activity assays
Antibody mediated complement activity was measured via MAC pore formation by monitoring fluorescence sulforhodamine B release on a CLARIOstar microplate reader (BMG Labtech, Offenburg, Germany) as previously described[27] using the CLARIOstar software (version 5.21 R2).

Briefly, purified sulforhodamine B liposomes displaying 1 mol% DNP as an antigen source were mixed with ice cold NHS (Complement Technologies, Tyler, TX, USA; 10% v/v final concentration) and fluorescence intensity was measured for 100 s at 21 °C before 50 nM IgG was added. Complement activation was assayed for 30 min. All experiments were performed in triplicate. Data were baseline corrected and analysed in GraphPad Prism (version 9.3.1).

## Polyacrylamide gel electrophoresis for mass spectrometry

Liposomes containing 1 mol% DNP were mixed with anti-DNP IgG3 (315 nM final concentration) and incubated at RT for 15 min. Next, samples were cooled down to 4 °C for 15 min. As a source of complement proteins, NHS (1% v/v final concentration) was added and incubated for 30 min at 4 °C. To purify liposome-IgG-C1-C4 complexes from unbound proteins, samples were centrifuged twice at 4 °C for 15 min at 20,000 $g$ using a Sigma 2-16k centrifuge (Sigma Laborzentrifugen GmbH, Harz, Germany), and the supernatant removed. The liposome pellet was resuspended in ice cold PBS and kept on ice till further use. Samples were diluted to equivalent concentrations in PBS and loaded, without heating denaturation, onto 4–12% pre-cast Bis-Tris Protein Gels in MOPS (pH 7.7) supplemented with 0.1% w/v SDS After running gels at 200 V at room temperature for 75 min, polyacrylamide gels were stained using SimplyBlue™ SafeStain for 20 min and destained in deionised water overnight. Gels were stored in 1% acetic acid at 4 °C before imaging using a white light conversion screen in a ChemiDoc XRS+.

## DNP-Biotin conjugation and purification

DNP-Biotin was produced via copper free click chemistry. Briefly, Biotin-PEG$_2$-Amine (10 mM final concentration, TCI Chemicals, Tokyo, Japan) and $N$-Succinimidyl $N$-(2,4-dinitrophenyl)-6-aminocaproate (DNP-NHS-ester; 12.5 mM final concentration, Sigma-Aldrich, St Louis, MO, USA) were mixed in PBS and DMSO at equimolar concentrations and incubated overnight at 37 °C with agitation at 1050 rpm in a ThermoMixer F2.0 (Eppendorf, Hamburg, Germany). Ion-pairing high-performance liquid chromatography (IP-HPLC) was used to purify conjugated product with 0.1 M triethylammonium acetate (TEAA; Sigma-Aldrich, St Louis, MO, USA) as an ion-pairing agent on a C18 coated stationary phase and acetonitrile (ACN; VWR, Radnor, Pennsylvania, USA). Samples were freeze-dried (CHRIST Beta 2-8 LSC basic Freeze Dryer, Martin Christ Gefriertrocknungsanlagen GmbH, Osterode am Harz, Germany) overnight and resuspended in nuclease free water. Purified DNP-Biotin concentrations were measured on a UV Vis spectrometry at 360 nm and calculated using the DNP-NHS extinction coefficient ($\varepsilon^{360nm}$) of 17,500 cm$^{-1}$ M$^{-1}$.

## Enzyme-linked immunosorbent assay (ELISA)

To detect complement protein binding to IgG subclasses, Maxisorp Nunc Immunoplates (Thermo Fisher Scientific, Waltham, Massachusetts, USA) were coated with 100 µl 10 µg/ml Streptavidin (Thermo Fisher Scientific, Waltham, Massachusetts, USA) in coating buffer (0.1 M Na$_2$CO$_3$, 0.1 M NaHCO$_3$, pH 9.6) for 1 h at 37 °C. After each incubation, plates were washed three times in PBS plus Tween-20 (0.05% w/v final concentration). Wells were blocked using 100 µl 3% bovine serum albumin (BSA) in PBS for 1 h at 37 °C. As an antigen source, 100 µl DNP-Biotin (2 nM final concentration) in 1% BSA in PBS was added and incubated for 1 h at 37 °C. Next, 50 µl anti-DNP IgG antibodies (final concentration from 0 to 150 nM) and NHS (1% v/v to detect C1q and MAC, or 0.5% to detect C4, final concentrations) diluted in RPMI 160 Medium (Thermo Fisher Scientific, Waltham, Massachusetts, USA) or RPMI alone were added to the wells and plates were incubated for 2 h at 37 °C. Afterwards, 50 µl primary antibody was added and incubated for 1 h at 37 °C. As a positive control, anti-DNP goat antibody (Bethyl Laboratories, Montgomery, Texas, USA) 1:1000 diluted in 1% BSA in PBS was added to a well with only RPMI. To detect

C1q, C4 and MAC, α-human C1q rabbit (1:1000 dilution; Dako, Denmark), α-human C4 goat (1:200,000 dilution; Complement Technology, Tyler, TX, USA), and α-human C5b-9 mouse (1:2500 dilution; Dako, Denmark) antibodies were used, respectively. All antibodies were diluted in 1% BSA in PBS. Next, 50 µl secondary antibodies conjugated to HRP were added and incubated for 1 h at 37 °C. As secondary antibodies, α-goat donkey with HRP (1:5000 dilution; Invitrogen, Thermo Fisher Scientific, Waltham, Massachusetts, USA), α-rabbit goat with HRP (1:2000 dilution; Dako, Denmark), and α-mouse goat with HRP (1:2000 dilution; Dako, Denmark) were used. Finally, 50 µl 2,2′-azino-bis(3-ethylbenzothiazoline-6-sulfonic acid) (2.5 mg/ml final concentration) in citric acid buffer (0.15 M, pH 4.2) with 0.015% (v/v) H$_2$O$_2$ were added and absorbance at 415 nm was measured after 15 min, 30 min and 45 min using a CLARIOstar microplate reader (BMG Labtech, Offenburg, Germany) using the CLARIOstar software (version 5.21 R2). Different complement protein detections were analysed separately in GraphPad Prism (version 9.3.1).

## Sample preparation for cryo-electron microscopy

Liposomes with 1 mol% DNP were incubated with anti-DNP IgG3 (315 nM final concentration) at room temperature for 20 min. Afterwards, the sample was cooled to 4 °C for 20 min, before either ice-cold PBS or NHS (1.5% v/v final concentration) was added, followed by another 20 min incubation on ice. Before vitrification, ice-cold bovine serum albumin-coated 5 nm gold colloids were added. Freshly plasma-cleaned 200 mesh copper grids, with lacey-carbon support (Electron Microscopy Sciences, PA, USA), were loaded into a Leica EMGP (Leica Microsystems, Wetzlar, Germany) and 6 µl sample was applied. Samples were incubated for 60 sec at 4 °C at 65% humidity, followed by 1.5 s blotting from the back and vitrification in liquid ethane. Grids were clipped and stored in liquid nitrogen until usage.

## Cryo-electron tomography data collection and reconstruction

For tilt series acquisition, a Talos Arctica (Thermo Fisher Scientific, Waltham, Massachusetts, USA) operating at 200 kV with a K3 direct electron detector and a Bioquantum energy filter (Gatan) was used. The energy filter was set to a slit width of 20 eV. Tilt series were acquired using Tomography 5 (version 5.5.0; Thermo Fisher Scientific, Waltham, Massachusetts, USA) in counting mode at ×49,000 magnification with a pixel size of 1.74 Å using a dose-symmetric scheme from 0 to ±57°, in 3° increments in a dose-symmetric tilt scheme[52]. Focusing between −3 µm and −6 µm and tracking were performed before each tilt acquisition. A total dose of 60 e$^-$/Å$^2$ was used.

Before reconstruction, raw frames were aligned using the "alignframes" command from IMOD (version 4.11)[53]. Tomograms were reconstructed after 2 × binning of tilt series using IMOD (version 4.11) with weighed back-projection. CTF estimation as well as correction and gold beat erasing was done in IMOD (version 4.11).

## Particle picking and subtomogram averaging

EMAN2 (version 2.91)[54] command e2spt_boxer_old.py was used to manually pick 1193 from 55 tomograms and 2561 particles from 101 tomograms for IgG3 and IgG3 plus NHS, respectively. Particles were extracted, with contrast inversion and normalisation, with a box size of 180 pixels for IgG3 alone, or 256 pixels for IgG3-C1 complexes, with a final pixel size of 3.48 Å/pixel.

See the Supplementary Methods for detailed routines used for subtomogram averaging of IgG3 and IgG3-C1-C4 complex. Briefly, for the reconstruction of the IgG3 Fc region, an initial model was created using a bilayer membrane model, which was filtered to 40 Å and white noise was added in EMAN2 (version 2.91) using e2proc3d.py. Further reconstructions were performed in Dynamo (version 1.1.157)[55]. The initial model was used as a template and particles were refined using multiple rounds of alignment and multireference classification to align all of the particles to the lipid bilayer. The resulting overall IgG3 map

was dominated by the Fab-membrane density (Supplementary Fig. 7). Multiple rounds of refinement and classification did not yield a high-resolution structure of the Fab array. To investigate the Fc platform, multireference classification and focussed refinement around the Fc region was performed, as described in the Supplementary Methods. One class yielded a clear Fc platform (Supplementary Fig. 7), which was then used for further gold-standard refinement. The particles were split into even-odd independent datasets and aligned to a low-resolution map of the IgG3-Fc region. To stop the Fc platform rotating away from the membrane, alignments were limited to ±10° out of plane, but otherwise particles were free to translate and rotate fully around the z axis (perpendicular to the membrane). Angular search and translation limits were reduced over 30 iterations to yield the non-symmetrized map at 19 Å (Supplementary Fig. 7). Clear C6 symmetry was present, and so the refinement was repeated with C6 symmetry imposed during refinement of the two half-datasets, yielding a final C6-symmetrized map at 14 Å (Supplementary Fig. 7).

To yield the IgG3-Fab array map, e2spt_boxer_old.py was used to manually pick 211 particles from the upper surface of the liposome shown in Supplementary Fig. 5c. Particles were extracted, with contrast inversion and normalisation, with a box size of 128 pixels and a final pixel size of 3.48 Å/pixel. An average of these particles (without any alignment) was used as an initial model. Next, 10 iterations of focussed alignment were performed using e2spt_classaverage.py, allowing full rotation around the z axis but only out-of-plane rotations up to 24° to stop particles misaligning with the known membrane orientation. Since this was just to align the particles visible in Supplementary Fig. 5c, no resolution estimation was performed.

For the reconstruction of the IgG3-C1-C4b complex, the EMAN2 command e2spt_binarytree.py was used to generate an initial model which was then lowpass filtered to 60 Å before subtomogram averaging using e2spt_classaverage.py from EMAN2 (version 2.91). Particles were split into even and odd datasets and all gold-standard procedures were followed. Particles were aligned over 20 iterations, with no symmetry applied and using a loose spherical mask centred on the Fc hexameric domain. The resulting map of the complete IgG3-C1-C4b complex reached 44 Å (model 1, Supplementary Fig. 10b,c).

For focussed refinement and classification of regions within the IgG3-C1-C4b complex, Dynamo (version 1.1.157)[55] was used. The initial model from above was refined using multiple rounds of alignment and multireference classification. Two of the resulting classes had clear density corresponding to C1 complexes bound to antibodies on a membrane. These classes were combined, yielding 2428 particles, which were then split into even-odd half datasets and used for gold-standard refinement to yield the overall IgG3-C1-C4b map (Fig. 3e and Supplementary Fig. 10b). Next, this map was subjected to focused refinements around the C1-Fc region, C1rs platform, and Fab-C4b region using the masks shown in Supplementary Fig. 10b. These focused refinements were all performed using the independently split even-odd half datasets aligned to the overall IgG3-C1-C4b map produced by Dynamo (version 1.1.157)[55].

## Model building

Fab domains with repeating motifs were identified from the protein database (PDB). Several common Fab-Fab interfaces were identified with the approximate distances observed here, which were; for the 3.8 nm repeat, 3WE6 and 5TDP; for the 5.25 nm repeat, 5TDP, 3HR5, 4JN2 and 5TDN (Supplementary Fig. 6a, b)[56,57]. We did not have sufficient resolution to identify light (L) and heavy (H) chains within each Fab, so we used the exemplary model for the interface 5TDP to model the Fab array due to the similarity between the model and map distances (Fig. 1d, f). In Supplementary Fig. 6c we show other possible combinations of crystal structures and how interactions would be possible combining structures from 5TDP, 3HR5 and 5TDN. These structures exist with various H-H, L-H and L-L interfaces that display

similar repeating distances[56,57], and higher resolution maps may be able to discriminate the exact orientation of the Fab domains within these arrays. IgG3 Fab domains were modelled using AlphaFold (version 2.2.2)[58], with the sequence for anti-DNP heavy and light chain variable domains[59], and placed onto the 5TDP array using UCSF Chimera (version 1.14)[60]. This was then briefly simulated using Isolde (version 1.5)[61] within UCSF ChimeraX (version 1.5)[62] to remove steric clashes to yield the model shown in Fig. 1d, f. In the same way IgG3 Fab domains were placed onto the array achieved from a combination of 5TDP, 3HR5 and 5TDN, which is shown in Supplementary Fig. 6c.

For the IgG3-Fc region, the PDB model with accession number 6D58 was used as a monomer. Multiple copies were placed in the C6-symmetrised IgG3-Fc map (Fig. 2a, b) in the same orientation as IgG1-Fc domains were observed in PDB model 1HZH[63]. The fit was then optimised, treating the domains as rigid bodies within UCSF Chimera (version 1.14) to form the hexameric IgG3-Fc platform. This was also briefly simulated using Isolde to remove steric clashes. To model the extended hinge regions, two polypeptide helices were constructed in UCSF Chimera (version 1.14) before disulfides were formed. This was allowed to relax by running a simulation in Isolde, although the exact structure of the hinge could not be determined from these data. Copies of this hinge polypeptide were placed between the Fab and Fc models, which were oriented to minimise the average hinge length, before peptide bonds were formed in UCSF Chimera (version 1.14) and simulated in Isolde (version 1.5). The relative positions of the Fc and Fab array could not be determined, and therefore other hinge topologies could be equally likely. The final IgG3 model is as shown in Fig. 2b.

The C1 complex was modelled as described in the Supplementary Methods and Supplementary Fig. 17 (See also reference [24]). For C4b, the PDB model with code 4XAM[23] was aligned to the PDB model with the code 5JPM, which displays the interaction of C4 with the complement protein MASP-2 (Supplementary Fig. 18)[64]. MASP-2 is homologous to C1s, which reveals the C4b CTC domain and TED domain are proximal and distal to C1s, respectively. This C4b orientation is in agreement with the structure of IgM-C1 interacting with C4b (EMD-4945), and as such the same starting orientation was used here. The fit of 4XAM was then refined by fitting into the subtomogram map of IgG3-C1-C4b as a rigid body using UCSF Chimera (version 1.14). Analysis of maps, models, and creation of figures was performed using UCSF ChimeraX (version 1.5)[62].

## In-gel tryptic digestion and LC-MS/MS analysis

Protein bands were subjected to reduction and alkylation (dithiothreitol and iodoacetamide) followed by in-gel trypsin (Worthington) digestion using a Proteineer DP digestion robot (Bruker Daltonics, Billerica, Massachusetts, USA). Peptides were extracted from the gel slices, lyophilised, dissolved in 100:0.1 water:formic acid (FA) v/v and subsequently analysed by on-line C18 nanoHPLC MS/MS with a system consisting of an Easy nLC 1000 gradient HPLC system (Thermo Fisher Scientific, Bremen, Germany), and an Orbitrap Fusion Lumos Tribrid mass spectrometer (Thermo Fisher Scientific, Bremen, Germany). Samples were injected onto a homemade precolumn (100 μm × 15 mm; Reprosil-Pur C18-AQ 3 μm, Dr. Maisch, Ammerbuch, Germany) and eluted via a homemade analytical nano-HPLC column (30 cm × 50 μm; Reprosil-Pur C18-AQ 3 μm). The gradient was run from 2 to 40% solvent B (20:80:0.1 water:acetonitrile:FA v/v/v) in 30 min. Solvent A was 100:0.1 water:FA v/v. The nano-HPLC column was drawn to a tip of ~5 μm and acted as the electrospray needle of the MS source.

The mass spectrometer was operated in data-dependent MS/MS mode with a cycle time of 3 s. In the master scan (MS1) the resolution was 120,000 and the scan range 400–1500, at an AGC target of 400,000 with maximum fill time of 50 ms. For MS2, precursors (charge stated 2–5) were selected using the quadrupole with an isolation width of 1.2 Th. Fragmentation was performed using HCD at a normalised collision energy at 32% and recording of the MS2 spectrum

in the Orbitrap. Dynamic exclusion after *n* = 1 was set, with an exclusion duration of 10 s.

IgG1 and IgG3 Fc *N*-glycopeptides and IgG3 *O*-glycopeptides were identified by manual inspection of the raw LC-MS/MS data of the corresponding tryptic peptides using XCalibur software (version 2.2 SP1.48; Thermo Fisher Scientific). To compare the overall *N*-glycan profile on the IgG1 and IgG3 antibodies used for our studies, summed MS-spectra covering the full width of the chromatographic peak of the tryptic peptides TKPREEQYN**S**T**Y**R (IgG1) and TKPREEQY**N**ST**F**R (IgG3) carrying different *N*-glycans were generated.

In a post-analysis process, raw data were first converted to peak lists using Proteome Discoverer (PD) (version 2.5.0.400; Thermo Fisher Scientific, Waltham, Massachusetts, USA). Subsequently, cross-linked peptides were identified using XlinkX (version 2.5)[65] as a node in PD. The database consisted of five proteins (IgG1/3 light and heavy chains, human C4b). Trypsin was selected as enzyme and two missed cleavages were allowed. Carbamidomethylation of cysteine and oxidation of methionine were selected as fixed and variable modifications, respectively. Precursor and fragment mass tolerance were set at 2 and 20 ppm, respectively. For the identification of thioester mediated crosslinked peptides, crosslinks between Q and S/T/K were defined (loss of NH3). Only crosslinked peptides between C4B and IgG1/3 with a score above 25 were selected for further analysis by manual interpretation of the MS/MS spectra.

### Reporting summary

Further information on research design is available in the Nature Portfolio Reporting Summary linked to this article.

## Data availability

The cryoEM maps and associated models generated for this study have been deposited in the EM database (EMDB) and protein database (PDB) with the following accession codes: IgG3-Fc hexamer map and model, PDB 8BTB and EMD-16227; IgG3-C1-C4b map, EMD-16241; focussed refinement of C1 region, EMD-16251; focussed refinement of C4b region, EMD-16250. Extracted particles of antigen-bound Ig3 and IgG3-C1 have been deposited in the Electron Microscopy Public Image Archive (EMPIAR)[66] with accession codes EMPIAR-11406 and EMPIAR-11407, respectively. Crosslinked peptides identified via LC-MS/MS analysis and XlinkX searches are supplied as Supplementary Data 1. The mass spectrometry proteomic data have been deposited to the ProteomeXchange Consortium via the Pride partner repository with the dataset identifier PXD039049[67]. Source data are provided with this paper.

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

## Acknowledgements

We thank Paul Parren for critically reviewing the paper and Yassene Mohammed for helping with the deposition of Mass spectrometry data to ProteomeXchange via the PRIDE database. This research was supported by the following grants to THS: European Research Council Grant 759517; The Netherlands Organization for Scientific Research Grants OCENW.KLEIN.291 and VI.Vidi.193.014. The mass spectrometry part was supported by the Dutch Research Council (NWO) Medium Investment Grant 91116004 (partly financed by ZonMw) to P.A.v.V.

## Author contributions

L.A. performed all assays and cryoET data collection. D.J.D. and L.A.T. constructed vectors and expressed and purified IgG. L.A. and T.H.S. performed tomogram reconstructions, subtomogram averaging and model building. P.A.v.V., R.T.N.T. and P.J.H. performed mass spectrometry data collection and analysis. L.A. and T.H.S. wrote the paper. All authors contributed to paper revisions. T.H.S. supervised the study.

## Competing interests

The authors declare no competing interests.
