## [Peer Review File · Nature Communications]

Complement is activated by elevated IgG3 hexameric platforms and deposits C4b onto distinct antibody domainsREVIEWER COMMENTS

Reviewer #1 (Remarks to the Author):

With fine molecular details, the present manuscript uncovers how immunoglobins of the IgG3 subclass can adopt unique conformations defining elevated Fc hexameric platforms which can activate the classical complement pathway. A sounded discussion further illustrates how extended hinges, a unique feature of this IgG subclass, can enhance pathogen neutralization, complement activation or receptor binding. Highly impressive cryoEM structural insights are provided onto such elevated macromolecular machineries. Complex sub-tomogram averaging has been performed to overcome lower resolution in EM maps associated to the enhanced flexibility of IgG3 (line 292). This has been achieved through multireference classification and focused refinement.

As compared to previous structural analyses of C1 complex with antigen-bound IgG1 or IgM, this study provides very original insights into Fab arrays, elevated hexameric Fc platforms which can be observed in absence of C1 (or hexamer stabilizing Fc-mutation), a different orientation of C4b (as compared to the one found with IgM/C1) and several well-defined locations of C4b covalent attachment onto IgG3 (finely defined using mass spectrometry analyses). A new orientation the C1s protease is also cleverly uncovered, which further fuels the ongoing discussion on how C1 activates, a challenging debate of the past decades!

These are fantastic and technically challenging new data!!!

Suggestions for improvement and questions:

1) Only three figures are used to illustrate the results plus one to illustrate the discussion. Clear figures are essential and this represents a real challenge here because of the wide range and complexity of the observations and discussion performed in this work. For the readers not fully familiar with IgG structures and subtypes, an illustration (in Figure S1 or an inset in Figure 1) showing the modular structure of IgG1 and IgG3 could help. This is also true for complement-mediated lysis and MAC which appears on Figure S1 without visual introduction.

2) Figure 1. The beginning of long hinges can be seen on the top of Fabs (in E or C): you might also indicate this feature in the figure legend. An additional inset with information displayed in Figure S5D, showing the averaged density of the 525 Fc platforms above Fab arrays would be a plus because the Fc details on Fig1B (or S3B) are really too thin! This would illustrate the text on lines 128-131. Readability would be improved if you adopt similar color codes for the different insets (C and E, F, and G).

3) Figure 2

C and D: The interpretative scheme is useful because the information is really difficult to see at some positions in the original figure displayed here.

E: What about the C1q stalk distortion (seen in previous complexes with C1 and antigen-bounded IgG1 or IgM) in these new complexes? Does the C1q stalk remain perpendicular to the Fc hexamer surface as shown in the model?

G and H: The 'new' C1s position looks on the other side of the C1q collagen stem. Right??? (details look so small). If yes, how could it swing from one position to the other? Or, if not, is it possible to suggest that C1s can swing from one position to the other on the outer side of the C1q cone? Where does the flexibility occur in C1s? (in CUB2 ??)

4) Figure 3. The figure is quite packed, especially for Inset C. However this remains readable at the cost of focused attention.

5) Figure S5

This figure S5 is really useful to fully integrate the message written in the main text. Part of it might be introduced in main figures.

6) On figure S8 (and Fig. 2) some C1 complexes look in close locations. Others do not. Could you estimate if the close proximity is a major trend or not throughout your images?

7) Complement is activated by elevated IgG3 hexameric platforms and deposit C4b onto distinct antibody domains (cf Fig.4c). Could you clarify for the reader that deposition of C4b on IgG is observed for different subtypes (cf Early observation of C4b deposition on IgG aggregates (PMID: 6906229) R D Campbell, A W Dodds, R R Porter, 1980). And that this does not prevent other C4b molecules to bind to the membrane surface (as shown in Fig. 4B), which is needed for lysis. Beyond this, mass spec analysis delivers new details that are really interesting and illustrate the variability and constraints associated to C4b binding. Any consequence on factor H or factor I binding to C4b if it is not on the membrane surface?

8) Averaged fixed positions are shown, but does membrane fluidity allow antigen-bound IgG3 displacements in the activation mechanism?? This likely needs to be suggested in the discussion about C1 activation.

9) Lines 296-305. Reorientation of C1s (and C1r?) is likely needed in the two activation hypotheses because of the constraints associated to the relative configuration required for C1r activating C1s (namely C1s activation site must be within C1r active site/S1 subsite). Could you show a supplementary

illustration or comment in the text how close these two sites are in Figure 4c? Among others, a question remains about how this C1r' gets activated. By C1r? These look very close! Another important question remains on how C1r and C1s could switch from one position to the next one during the activating cycle?

10) This study is performed using very small DNP antigens which can diffuse at the surface of the liposome fluid membrane. How the current observations could be extended to larger antigens?

Last minor observations:

Lines 132-133 'this is the first time such a hexagonal Fc platform has been observed in the absence of the C1 complex'. May be add 'in absence of mutations enhancing Fc-hexamer formation?

Sentence in lines 223-225 not 100% clear, because of only one threonine in the sequence stretch

Typing error in line 210

Reviewer #2 (Remarks to the Author):

This interesting study brings together a variety of experimental approaches and informed interpretation to generate novel models of IgG3 binding and complement activation. Taken as a whole the paper is interesting and provides answers to many past questions.

My major comment is that the figures are complex and the figure legends do not always provide sufficient information for the naïve reader to understand exactly what they are being shown. A careful reading of these to aid clarity would be appreciated.

The subtomograms being fit are, unsurprisingly, for small and mostly asymmetric objects, low resolution. No data are provided to give a numerical sense of the quality of fit and the text does not really hint at the limitations imposed on the models of the ambiguities that fitting at such low resolution leaves. These are not comments intended to prevent publication - but this referee feels it would be important for some fit metrics to be presented - perhaps correlation for preferred fit versus correlation for next best fit? along with a more explicit acknowledgement of the limitations of the study imposed by resolution. The use of MS data to further inform is a strength and this referee has no doubt the models generated are likely true representations of the object.

With a thoughtful revisiting of the text to address these points, the manuscript will be suitable for publication and will be of interest to a wide audience.

Reviewer #3 (Remarks to the Author):

Technical review electron microscopy

The samples and images appear clean and the features described are clearly visible.

I have significant concerns about the image analysis, the description of the image analysis, and the model building, in particular:

Most of the image processing has not been performed using a “gold standard” approach – instead the data has been globally aligned, and only divided into even and odd datasets late in the processing pipelines. Such an approach is only acceptable when careful documentation of filtering parameters confirms there is no noise alignment or reference bias (for example if information at better than 40Å is not considered at any time in the global processing) but that is not the case here. It is therefore not currently possible to assess if the final structures are correct or not. In a challenging case such as this, validating the structure is best achieved by performing all processing on independent half datasets from the beginning.

Figure S5D does not convincingly show that there is a hexamer. The image processing should be on independent half datasets and there should be a proper assessment of whether the C1 structure has six-fold features or not

It is essential to include a quantitative assessment/validation of the final models – how well do the structures fit the densities, how many alternative arrangements of proteins would be consistent with the density, are models presented the only solutions or one of many solutions consistent with the data?

The 5.1-5.4 nm repeat is not clear in S4D. The images show repeats, but not the power spectrum. It should be possible to improve the analysis to show this repeat clearly if it is real, perhaps using some patch alignment.

There seems to be an assumption that the repeat distance corresponds to the protein size, but this depends upon how the protein is arranged to form a repeating arrangement.

The image processing and model building sections of the methods should be much more detailed.

Standard tables for EM data collection and image processing need to be included.

Reviewer #4 (Remarks to the Author):

In the manuscript titled “Complement is activated by elevated IgG3 hexameric platforms and deposits C4b onto distinct antibody domains” by Abendstein et al. the authors have attempted to investigate the structure of antigen bound IgG3 and complexed with complement components, by a combination of biophysical assays including mass spectrometry. This ambitious study undertaken by the authors provides important structural clues on how antigen is bound to IgG3 and IgG1. Though more work is needed to completely untangle the structure, I think this study is a start in the right direction. The idea sounds fundamentally sound to me but because my expertise is in mass spectrometry, I feel most comfortable reviewing the mass spectrometry part of the manuscript. The manuscript is mostly well-written though there are some parts specifically regarding mass spectrometry that need to be addressed (see comments below). Once these are addressed, I think that this manuscript can be published.

Specific Comments & Questions:

1) I suggest that authors define some abbreviations like Fab, Fc, C1, C1q, SEC (in Figure S1) when used for the 1st time in the manuscript, for the readers.

2) I suggest that authors provide citations for line 36 on page# 2 “IgG subclasses are highly similar, sharing ~95% sequence homology in the heavy chain constant domains, and all contain a Fc domain linked to two antigen-binding Fab domains via a disulfide-linked polyproline hinge”. Same with the line “IgG1, IgG2 and IgG4 are structurally homologous, with relatively short and rigid hinge regions containing 2-4 disulfide bonds, which limit the flexibility and size of the antibody complex domains via a disulfide-linked polyproline hinge” – which needs a reference.

3) Though cryo-EM / crystallography is not my expertise, I am curious to know how confident the authors are of the distances measured like: 22nm by which Fc platform is modeled to rise away from the antigens, as in these are absolute determinations, how big or small is the error bar or standard error on these?

4) In Figure 4B, I am unable to understand what the authors mean by “Below are height comparisons of various complement components with IgG3 and IgG1”. Please clarify.

5) How did the authors decide on performing HCD with a normalized collision energy of 32%, as in how was this energy selected? How did the authors decide to perform HCD and not ETD for N-linked glycan analysis.

6) Authors have performed glycoproteomics, but have not mentioned how they characterized the N-linked and O-linked glycans by mass spectrometry? What software was used for glycan analysis? How were the O-linked glycans/ glycopeptides characterized (any protease used to chop up the glycopeptides?) and identified? Was any database used?

7) How did the authors confirm the identify of N-linked glycans and their isomers? Was MS/MS used for this purpose- assigning N-linked glycan isomers can quite challenging.

8) In FigureS13, how did the authors confirm the presence of Neu5Ac and that it is not Neu5Gc?

9) How do the results of this study compare with Klaus, T., Bzowska, M., Kulesza, M. et al. Agglutinating mouse IgG3 compares favourably with IgMs in typing of the blood group B antigen: Functionality and stability studies. *Sci Rep* 6, 30938 (2016). <https://doi.org/10.1038/srep30938> ?

10) In figure 3, the peptide SCDTPPPCPR is observed to be eluted at 2 distinct retention times, do the authors have a handle on what is causing this? Probably different conformation when linked to C4b?

11) I am curious to know if the authors considered using SEC-MALS to gain insights in to the binding mechanism/ kinetics of IgG3 to antigen and complexed with complement components- I am in no way suggesting that they do that now.

We thank the reviewers for performing such a thorough and helpful critique of our manuscript. We have incorporated their comments and suggestions and believe that they have improved the manuscript. Below, we address each of their comments point-by-point, linking the numbered changes made to the new version of the manuscript.

Reviewer #1:

We thank the reviewer for their kind words and are glad they find our work “fantastic and technically challenging”.

1.1) Only three figures are used to illustrate the results plus one to illustrate the discussion. Clear figures are essential and this represents a real challenge here because of the wide range and complexity of the observations and discussion performed in this work. For the readers not fully familiar with IgG structures and subtypes, an illustration (in Figure S1 or an inset in Figure 1) showing the modular structure of IgG1 and IgG3 could help. This is also true for complement-mediated lysis and MAC which appears on Figure S1 without visual introduction.

*Thank you for the suggestion. We have now split Figs 1 and 4 into separate halves, so there are now 6 figures in total that have also been enlarged to make the details clearer. We have also added a new **Fig. S1** to describe each of the proteins mentioned in the introduction, which acts as a resource for the rest of the manuscript too.*

1.2a) Figure 1. The beginning of long hinges can be seen on the top of Fabs (in E or C): you might also indicate this feature in the figure legend.

*Label added to **Fig 1c** (e has been changed so no longer needs the label).*

1.2b) An additional inset with information displayed in Figure S5D, showing the averaged density of the 525 Fc platforms above Fab arrays would be a plus because the Fc details on Fig1B (or S3B) are really too thin! This would illustrate the text on lines 128-131.

*We have altered the text on lines 131 - 136 to make this section clearer; these 525 particles were not processed separately, instead, they were included in the 1193 particles used for subtomogram averaging (see **Fig. S6**, old **Fig. S5**).*

*Furthermore, we have made the figures larger (**Figs. 3 and S6**) to make the details easier to see and understand our text on lines 131 - 136 (before lines 128-131).*

1.2c) Readability would be improved if you adopt similar color codes for the different insets (C and E, F, and G).

*We have changed the colours in **Fig 1b** and **d** (new labels) to clarify which density corresponds to Fab or membrane regions: violet and green now represent IgG3-Fabs, whilst cyan represents lipid bilayer. We have split Fig 1 so it is now Figs 1 and 2, and the colours have been modified to make identification clearer.*

1.3a) Figure 2

C and D: The interpretative scheme is useful because the information is really difficult to see at some positions in the original figure displayed here. E: What about the C1q stalk distortion (seen in previous complexes with C1 and antigen-bounded IgG1 or IgM) in these new complexes? Does the C1q stalk remain perpendicular to the Fc hexamer surface as shown in the model?

Thank you for the suggestion to add this information. We have added a comparison to **Figure S10c** showing that the stalk is much more upright than in IgM-C1 and IgG1-C1 structures. We have also added the following text to the results and discussion sections.

Lines 171-172: "In contrast to IgG1 and IgM, the C1q stalk was more upright on IgG3 (**Fig. S10c**)."

Line 304: "..., although with a more upright stalk (**Fig. S10c**).

Line 319-322: "Furthermore, the more upright C1q stalk observed here, compared to structures of IgG1 and IgM (**Fig. S10c**), may indicate that the stalk tilts less during C1r/s activation than after C1s cleavage or during C4b deposition"

1.3b)

G and H: The 'new' C1s position looks on the other side of the C1q collagen stem. Right??? (details look so small). If yes, how could it swing from one position to the other? Or, if not, is it possible to suggest that C1s can swing from one position to the other on the outer side of the C1q cone? Where does the flexibility occur in C1s? (in CUB2 ??)

We have enlarged the figure to make this conformational change and location more clear. C1s does not move to the other side of the C1q collagen arm, but can rotate around the C1q collagen arm at the CUB2 region, as there is no calcium binding CUB2 to C1q at this position. We have added these comments and references to the manuscript on lines 308-312:

"This flexibility is possible for C1s due to the absence of collagen-binding between C1s-CUB2 and C1q at this location [**REF 44**], which is in contrast to the C1q collagen-binding sites found C1r-CUB1 and CUB2, and C1s-CUB1, which are observed to be less structurally heterogeneous. This C1s flexibility is reflected in the variable C1s configurations previously observed [**REF 24, 45, 46**]."

1.4) Figure 3. The figure is quite packed, especially for Inset C. However this remains readable at the cost of focused attention.

We have added circles to **Figure 4c** (old figure 3) to highlight the sites of covalent attachment to clarify the main point of this panel.

1.5) Figure S5

This figure S5 is really useful to fully integrate the message written in the main text. Part of it might be introduced in main figures.

We thank the reviewer for this suggestion and have now added this to (new) **Figure 2c and d** to help the narrative of the paper.

1.6) On figure S8 (and Fig. 2) some C1 complexes look in close locations. Others do not. Could you estimate if the close proximity is a major trend or not throughout your images?

This is an interesting suggestion that we did implement, revealing a particle-particle distance mode of around 25 nm (see figure below). Although for a previous paper on IgG1-C1 complexes we interpreted a similar distance (23 nm) to be due to C1-C1 interactions (**REF 22** in the manuscript), here, due to the clustering induced by IgG3-Fab domains, we did not feel we had enough data to address the cause of this clustering so limited our discussion to the following in lines 189-191:

"Empty density was also located adjacent to the IgG3-Fc hexameric ring (white arrow, **Fig. 3g**), which may be due to neighbouring IgG3 complexes, as previously observed for IgG1"

1.7) Complement is activated by elevated IgG3 hexameric platforms and deposit C4b onto distinct antibody domains (cf Fig.4c). Could you clarify for the reader that deposition of C4b on IgG is observed for different subtypes (cf Early observation of C4b deposition on IgG aggregates (PMID: 6906229) R D Campbell, A W Dodds, R R Porter, 1980). And that this does not prevent other C4b molecules to bind to the membrane surface (as shown in Fig. 4B), which is needed for lysis. Beyond this, mass spec analysis delivers new details that are really interesting and illustrate the variability and constraints associated to C4b binding. Any consequence on factor H or factor I binding to C4b if it is not on the membrane surface?

We have added these citations and expanded the discussion on lines 335-343 to address C4b binding to IgG isotypes:

“Previous studies have also described C4b binding to IgG aggregates, although the exact binding site of C4b to these IgG aggregates could not be localised [REF 48, 49]. However, by using cryoEM C4b was previously observed being deposited onto lipid membranes by IgM-C1 complexes [REF 24], but has not been seen in complexes of IgG1-C1 [REF 22]. Although complement progression indicates that C4b is also deposited on to lipid membranes, allowing formation of C3 convertases (Figs. S2c, d), the height of IgG3-C1 meant that the lipid bilayer was further from C1s than the length of C4b (~18 nm and ~14 nm, respectively), and consequently C4b was deposited onto one of three sites on IgG3 as well.”

We have added a brief discussion on complement regulators to lines 364-369:

“How this C4b location affects complement regulation remains to be determined. Membrane-bound complement regulators, such as membrane cofactor protein (MCP) [REF 25], may not be as effective when C4b is not colocalised on the membrane. In contrast, soluble regulators, such as C4b-binding protein (C4BP) and Factor I, or extended molecules, such as CR1 [REF 25], may have more access to IgG3-associated C4b.”

1.8) Averaged fixed positions are shown, but does membrane fluidity allow antigen-bound IgG3 displacements in the activation mechanism?? This likely needs to be suggested in the discussion about C1 activation.

As suggested, we have addressed this comment in the discussion on lines 329-334:

“Cross activation would rely on C1 complexes binding close to one another or, as in this case, to antibody-antigen complexes that are able to diffuse laterally in the fluid lipid bilayer. Membrane fluidity and antigen diffusion was proposed to allow antibody rearrangement to allow C4b deposition from IgM-C1 complexes [REF 24]. Whether such a mechanism is also necessary or applicable here could not be determined from structural data. However, the

extended hinge of IgG3 caused deposition of C4b by C1s onto the antibody itself (Figs. 4a & S10e)."

1.9) Lines 296-305. Reorientation of C1s (and C1r?) is likely needed in the two activation hypotheses because of the constraints associated to the relative configuration required for C1r activating C1s (namely C1s activation site must be within C1r active site/S1 subsite). Could you show a supplementary illustration or comment in the text how close these two sites are in Figure 4c? Among others, a question remains about how this C1r' gets activated. By C1r? These look very close! Another important question remains on how C1r and C1s could switch from one position to the next one during the activating cycle?

Our response to 1.3b (above) addresses this comment. Furthermore, we have now included measurements in Fig. 3h, and added a new figure (Fig. 6a) to enlarge this view.

1.10) This study is performed using very small DNP antigens which can diffuse at the surface of the liposome fluid membrane. How the current observations could be extended to larger antigens?

We have addressed this point in response 1.8 above, and also added more detail in the discussion on lines 373-378:

"... C4b deposition and therefore compromised complement activation. This could also depend on the location and environment of the antigen, and there is evidence that different antigens may have different ideal hinge lengths for C4b deposition [REF 17, 50], which can now be explained by our structural data. It will be intriguing determine whether there is a correlation between antigen size and epitope location with complement activation by antigen subclass or IgG3 allotype."

1.11) Last minor observations:

Lines 132-133 'this is the first time such a hexagonal Fc platform has been observed in the absence of the C1 complex'. May be add 'in absence of mutations enhancing Fc-hexamer formation'?

We have added the following text to address this on line 138:

"...although this is the first time such a hexagonal Fc platform has been observed for non-mutated or antigen-bound antibodies in the absence of the C1 complex."

1.12) Sentence in lines 223-225 not 100% clear, because of only one threonine in the sequence stretch

We have modified this and the previous sentence to read on lines 229-232:

"Within this triple repeating hinge sequence, three peptides could be identified by MS, with either the serine (peak at 31.31 min) or the threonine (peak at 29.49 min) as the site of C4b covalent attachment to SCDTPPPCPR (Figs. 4d & S11b)..."

And on lines 233-236:

"These same threonine residues in the triple-repeat IgG3 hinge region sequences are also the location of the O-linked glycans, which occupy ~10% of the six available sites (Figs. S1a) [REF 8], and were also detected in our IgG3 (Fig. S14)."

1.13) Typing error in line 210

Corrected

Reviewer #2:

We are glad the reviewer found our work interesting and thank the reviewer for their robust critique. We have addressed their concerns below.

2.1) My major comment is that the figures are complex and the figure legends do not always provide sufficient information for the naïve reader to understand exactly what they are being shown. A careful reading of these to aid clarity would be appreciated.

Thank you for this suggestion, which was also suggested by reviewer #1. We have modified all of the main-text figures and legends, as well as many of the supplemental figures, to aid visualization and interpretation. This has increased the number of figures to 6, which has also allowed us to make the legends more descriptive. Please also see responses above and the tracked-changes in the attached manuscript.

2.2) The subtomograms being fit are, unsurprisingly, for small and mostly asymmetric objects, low resolution. No data are provided to give a numerical sense of the quality of fit and the text does not really hint at the limitations imposed on the models of the ambiguities that fitting at such low resolution leaves. These are not comments intended to prevent publication - but this referee feels it would be important for some fit metrics to be presented - perhaps correlation for preferred fit versus correlation for next best fit? along with a more explicit acknowledgement of the limitations of the study imposed by resolution. The use of MS data to further inform is a strength and this referee has no doubt the models generated are likely true representations of the object.

*We thank the reviewer for their suggestions. We have added **Table S3**, to provide a better overview about how data were collected. Additionally, we extended the Methods section for subtomogram averaging, which can now be found in the supplementary material. For the model building sections in the main text and supplemental methods, we have added numerous comments about the limitations placed on the model building by the resolution of our maps. Please see lines 110-122, 139-145, 159-207, 516-570. Rather than build multiple models and calculate their CCCs, this instead has allowed us to use published data to make assumptions on protein domain orientations where the exact structure was not resolvable.*

With a thoughtful revisiting of the text to address these points, the manuscript will be suitable for publication and will be of interest to a wide audience.

Reviewer #3:

I have significant concerns about the image analysis, the description of the image analysis, and the model building, in particular:

We thank the reviewer for their detailed critique of our data and analyses. We have added an extensive methods section to the SI, with reference to other papers that also deal with the unusual methods required for structural biology of low-resolution membrane-bound complexes, which we hope has satisfied the reviewer's main concerns. We have also performed additional analysis suggested by the reviewer on the Fab array, which yielded some highly interesting results and that we describe below, along with addressing all of their points raised.

3.1) Most of the image processing has not been performed using a “gold standard” approach – instead the data has been globally aligned, and only divided into even and odd datasets late in the processing pipelines. Such an approach is only acceptable when careful documentation of filtering parameters confirms there is no noise alignment or reference bias (for example if information at better than 40Å is not considered at any time in the global processing) but that is not the case here. It is therefore not currently possible to assess if the final structures are correct or not. In a challenging case such as this, validating the structure is best achieved by performing all processing on independent half datasets from the beginning.

*We had unintentionally omitted the methods sections for the map reconstructed using EMAN2, shown in **Fig. S9b** (Model 1), which was indeed achieved using gold-standard procedures: Before aligning to the initial map, which was filtered to 60 Å, the particles were split into e/o datasets and refined completely independently to reach the 44 Å reported resolution. We have now added this to the methods section.*

Due to this comparatively poor achievable resolution, which we posit is due to the independent flexibility of the IgG3, C1q and C1rs enzymes (as we discuss on lines 159-165), we decided to realign all particles using Dynamo for downstream classification and focussed refinement. We have now added a comprehensive methods section describing all of the steps required in the supplemental methods. Although this was indeed not performed using gold-standard procedures, this is not unusual for samples such as these; since these particles are all bound to a lipid membrane, the orientation of the particles is not completely random, as it would be for ideal solubilised complexes. Therefore, some subjectivity is useful in the early rounds to ensure all particles are oriented correctly with respect to the lipid membrane, which removes some degrees of rotational freedom that is inherent to the gold-standard processing pipeline.

This approach is well-established for low-resolution and membrane-associated particles, see, e.g., the following references:

- Burt, A. et al. Complete structure of the chemosensory array core signalling unit in an *E. coli* minicell strain. *Nature Communications* **11**, 743 (2020).
- Wolff, G. et al. A molecular pore spans the double membrane of the coronavirus replication organelle. *Science* **369**, 1395-1398 (2020)
- Dahmane, S. et al. Membrane-assisted assembly and selective secretory autophagy of enteroviruses. *Nature Communications* **13**, 59-89 (2022).
- Navarro, P. P. et al. Protocols for Subtomogram Averaging of Membrane Proteins in the Dynamo Software Package. *Frontiers in Molecular Biosciences* **5**, 82 (2018).

*Nevertheless, as we now describe in detail in the methods section, the particles were indeed split into half datasets, and these datasets were maintained throughout the processing (particles did not swap datasets). We have adjusted **Fig. S9b** to depict this in a clearer fashion. In order to remove remaining concerns, we have uploaded the particles extracted from the tomograms to EMPIAR under deposition code EMPIAR-11406 (IgG3) and EMPIAR-11407 (IgG3-C1-C4b), which will be accessible after publishing. These are non-aligned and will allow others to replicate our results.*

3.2) Figure S5D does not convincingly show that there is a hexamer. The image processing should be on independent half datasets and there should be a proper assessment of whether the C1 structure has six-fold features or not

*Please see response 3.1 above regarding independent e/o datasets, which were indeed maintained for processing. We have added a slice through the density map in **Fig. S6a** to aid interpretation of this volume, which more clearly shows the hexameric nature of the Fc platform which was not previously apparent in the isosurface depiction. We also determined the rotational correlation coefficients for the slices shown and added the graph to **Fig. S6b**, which shows 6 peaks, in agreement with C6 symmetry.*

3.3) It is essential to include a quantitative assessment/validation of the final models – how well do the structures fit the densities, how many alternative arrangements of proteins would be consistent with the density, are models presented the only solutions or one of many solutions consistent with the data?

*We have added a table into the supplemental information (**Table S3**) including parameters conventional for these types of low-resolution data. We have also added into the model building section of the supplementary methods that includes discussions on the limitations placed on our model by the map resolution, please see lines 110-122, 139-145, 159-207, 516-570.*

3.4) The 5.1-5.4 nm repeat is not clear in S4D. The images show repeats, but not the power spectrum. It should be possible to improve the analysis to show this repeat clearly if it is real, perhaps using some patch alignment.

*This excellent suggestion stimulated us to perform additional subtomogram averaging of the data presented in **Fig. S5d** (old figure Fig. S4D), which revealed much clearer Fab-Fab interfaces. We combined this new map with data mining of the PDB to identify similar interfaces present in existing crystal structures that, to our knowledge, have not been previously remarked upon as existing in native environments, as described here. This discovery necessitated a rewrite of the relevant sections, which we have highlighted in the manuscript on lines 113-116 and 118-122, and an addition to the supplementary methods section.*

3.5) There seems to be an assumption that the repeat distance corresponds to the protein size, but this depends upon how the protein is arranged to form a repeating arrangement.

*Our new subtomogram analysis, described in response 3.4 above, has revealed much clearer Fab-Fab interfaces, which allowed us to build a much more faithful model explaining these data. This model building, and limitations thereof, is described in the new supplementary methods section, and presented in new **Figures 1b,c** and **S5d**.*

3.6) The image processing and model building sections of the methods should be much more detailed.

To aid understanding of our approach and routines, we have greatly expanded the methods section to include details about subtomogram averaging and model construction, including a discussion of the limitations that our map imposed. We have also deposited the extracted particles for IgG3 and IgG3-C1 complexes to the EMPIAR open-access database. This is noted in the new “Data Availability” statement, along with the relevant EMDB and PDB codes of the deposited maps and model.

3.7) Standard tables for EM data collection and image processing need to be included.

*We have now added these data in Supplemental **Table S3**. See also response 3.3.*

Reviewer #4:

We thank the reviewer for their helpful evaluation of our manuscript. Below we have answered all questions and comments and have incorporated these into our manuscript. We are also happy to hear that the reviewer states that the “manuscript can be published”.

4.1) I suggest that authors define some abbreviations like Fab, Fc, C1, C1q, SEC (in Figure S1) when used for the 1st time in the manuscript, for the readers.

*This is a helpful suggestion that we have enacted: We have included the abbreviations for Fab, Fc, C1 and SEC in either the introduction or, in case of SEC, below the figure. We have also added **Fig. S1** to describe each of the proteins mentioned in the introduction, as also suggested by Reviewer 1 (see also response 1.1 above).*

4.2) I suggest that authors provide citations for line 36 n page# 2 “IgG subclasses are highly similar, sharing ~95% sequence homology in the heavy chain constant domains, and all contain a Fc domain linked to two antigen-binding Fab domains via a disulfide-linked polyproline hinge”. Same with the line “IgG1, IgG2 and IgG4 are structurally homologous, with relatively short and rigid hinge regions containing 2-4 disulfide bonds, which limit the flexibility and size of the antibody complex domains via a disulfide-linked polyproline hinge” – which needs a reference.

We included the missing references to these sentences.

*On line 36-38: “IgG subclasses are highly similar, sharing ~95% sequence homology in the heavy chain constant domains, and all contain a Fc domain linked to two antigen-binding Fab domains via a disulfide-linked polyproline hinge”: **REF 1, 2, 3 and 4***

*On line 38-41: “IgG1, IgG2 and IgG4 are structurally homologous, with relatively short and rigid hinge regions containing 2-4 disulfide bonds, which limit the flexibility and size of the antibody complex (Fig. S1a)”: **REF 2, 4, 5, 6 and 7***

4.3) Though cryo-EM / crystallography is not my expertise, I am curious to know how confident the authors are of the distances measured like: 22nm by which Fc platform is modeled to rise away from the antigens, as in these are absolute determinations, how big or small is the error bar or standard error on these?

We calculated the standard deviation. It is 3.5 nm and was mentioned in the results part (line 127). To avoid confusion we also now include this number in the discussion section (line 282)

4.4) In Figure 4B, I am unable to understand what the authors mean by “Below are height comparisons of various complement components with IgG3 and IgG1”. Please clarify.

We have enlarged this figure and changed the legend to make this statement clearer, and added the following text to the discussion:

“Modelling this arrangement reveals how IgG1 hexamers are found below the height of opsonizing complement components (Fig. 5c), whilst IgG3 is above these proteins, and therefore more exposed for binding.”

4.5) How did the authors decide on performing HCD with a normalized collision energy of 32%, as in how was this energy selected?

These are our standard setting for the analysis of tryptic peptides in our bottom-up proteomics experiments using our Orbitrap Fusion LUMOS. It works well for a broad range of different peptides and during the course of our experiments there was no reason to adjust if for our analyses of cross-linked or glycopeptides.

How did the authors decide to perform HCD and not ETD for N-linked glycan analysis.

ETD is particularly useful for site-assignment but given the conserved N-glycosylation site in the IgG Fc part, this was not necessary for our N-glycopeptide analyses.

4.6) Authors have performed glycoproteomics, but have not mentioned how they characterized the N-linked and O-linked glycans by mass spectrometry? What software was used for glycan analysis? How were the O-linked glycans/ glycopeptides characterized (any protease used to chop up the glycopeptides?) and identified? Was any database used?

We apologize for not providing this information. Glycopeptides were manually annotated using the raw data in the XCalibur software (Thermo). Therefore, we have not used any search algorithm or database. For the Fc N-glycopeptides, we performed standard in-gel tryptic digestion of the IgG1 and IgG3 bands shown on the SDS-PAGE gel of Figure S11A. The IgG3 O-glycopeptide (Fig. S14, identified in the C4b-IgG3 band (lane "Lipos+IgG3+NHS" on the SDS-PAGE gel of Fig. S11a)) was also identified using our standard procedure (trypsin digestion and LC-MS/MS analysis). Therefore, we have not performed any additional enzymatic treatment (for example to remove the glycans) or used specific settings during the LC-MS/MS analyses. We have now added this information to the Materials and Methods section on lines 591-596. "IgG1 and IgG3 Fc N-glycopeptides and IgG3 O-glycopeptides were identified by manual inspection of the raw LC-MS/MS data of the corresponding tryptic peptides using XCalibur software (Thermo). To compare the overall N-glycan profile on the IgG1 and IgG3 antibodies used for our studies, summed MS-spectra covering the full width of the chromatographic peak of the tryptic peptides TKPREEQYNSTYR (IgG1) and TKPREEQYNSTFR (IgG3) carrying different N-glycans were generated."

4.7) How did the authors confirm the identify of N-linked glycans and their isomers? Was MS/MS used for this purpose- assigning N-linked glycan isomers can quite challenging.

Indeed, during the data dependent LC-MS/MS analysis fragmentation of the glycopeptides was performed. For the reviewer, we here provide the MS/MS spectra of the major IgG1 and IgG3 Fc glycopeptides:

Figure 1: HCD MS/MS spectra of the IgG1 (upper part) and IgG3 (lower part) Fc tryptic peptides carrying an H3N4F1 N-glycan. Pep=peptide.

Note the difference in the mass of the peaks (indicated with the two-headed arrows) due to the small difference in the peptide sequence (Y vs F). In the manuscript, we provide summed MS-spectra (covering the full width of the chromatographic peak) of the major IgG1 and IgG3 Fc N-glycopeptides (Fig. S3), in order to visualize the overall glycosylation pattern on the IgG1 and IgG3 as used in our experiments. The structural annotation was based on canonical N-glycosylation core structures and what is known about Fc-N-glycosylation, a major topic in our group.

4.8) In FigureS13, how did the authors confirm the presence of Neu5Ac and that it is not Neu5Gc?

The Neu5Ac specific oxonium ion was observed at m/z 292.102 (and minus water at 274.091). Neu5Gc has a different mass. The typical sialic acid in human glycoproteins is Neu5Ac. The antibodies used for our experiments were produced in human cells (Expi293F) and humans cannot produce Neu5Gc.

4.9) How do the results of this study compare with Klaus, T., Bzowska, M., Kulesza, M. et al. Agglutinating mouse IgG3 compares favourably with IgMs in typing of the blood group B antigen: Functionality and stability studies. Sci Rep 6, 30938 (2016). <https://doi.org/10.1038/srep30938> ?

Unfortunately, mouse and human antibodies do not follow the same nomenclature. Therefore, mouse IgG3 is not similar to human IgG3 and comparisons between these two antibodies would not be reasonable.

4.10) In figure 3, the peptide SCDTPPPPCR is observed to be eluted at 2 distinct retention times, do the authors have a handle on what is causing this? Probably different conformation when linked to C4b?

*Indeed, the SCDTPPPCPR cross-linked to the C4b peptide GCGEQTmIYLAPTLAASR peptide eluted at two different retention times in our chromatography setup. Based on the MS/MS data as shown in **Fig. S11b** we could show that it depends on how it is cross-linked to the glutamine in the C4b peptide. The peak at 29.49 min in **Fig. 3b** corresponds to linkage of the glutamine to Thr-4 in the SCDTPPPCPR peptide, while the peak at 31.31 min corresponds to linkage to Ser-1 within this peptide.*

4.11) I am curious to know if the authors considered using SEC-MALS to gain insights in to the binding mechanism/ kinetics of IgG3 to antigen and complexed with complement components- I am in no way suggesting that they do that now.

This is a thought-provoking question but one that we have not yet addressed, but we thank the reviewer for their interest!

REVIEWER COMMENTS

Reviewer #1 (Remarks to the Author):

The new revised version is improved and my questions have mainly been addressed.

Congratulations for this beautiful work!

The modularity and combined flexibility and rigidity of IgG3 hinges is really amazing!

Minor concerns are the following:

Further attention is required for a correct use of labels in the Figure 1 legend, as well as for the one used in line 122.

I could not see the end of the Figure 3 legend.

I was also quite surprised by the line 320 added in the discussion, but this is discussion.

Reviewer #3 (Remarks to the Author):

The revised manuscript is much improved, but not all reviewer comments from the original submission are satisfactorily addressed

The methods are not completely clear, but the data in figures 2 and S6 seem not to have been aligned using independent half datasets, and the authors still use a global alignment step in Dynamo in figure S9. Limiting some degrees of freedom using the adjacent bilayer does not alter the fact that splitting the dataset into two halves for independent processing avoids many possible processing pitfalls, gives confidence in the result and allows a clear assessment of resolution. There are many possible pitfalls when aligning flexible and variable objects from small datasets with small masks. In my opinion authors should have enough confidence in the quality of the data to perform all processing using two independent half datasets.

The authors still do not present data that give a numerical sense of the quality of fitting during model building, and it is difficult to assess how much more likely the presented model is as other alternative possible models. The revised text is improved but, in my opinion, it is still too difficult for the reader to assess whether there are any other qualitatively different models that may also be consistent with the data. For example, the methods section suggests that there are multiple possible arrangements of the Fab array, one of which is shown in 1f - are the other arrangements equally likely? For example, the authors state that C4b clearly fits into the density – how does the presented fit compare to the next best fit? For example, 1PK6 is fitted as a rigid body, but then 6FCZ is used to define the orientation – what does this mean? When comparing PDB to density, the authors should illustrate or describe whether the solution is unique, or if not, whether the next best solution is also reasonable.

There is no legend for panel 1f.

Reviewer #4 (Remarks to the Author):

The authors have addressed my comments and questions adequately & therefore I recommend the updated version of the manuscript for publication.

We thank all reviewers for their helpful comments and suggestion.

Below, we address each of their comments point-by-point, linking the numbered changes made to the new version of the manuscript.

Reviewer #1:

The new revised version is improved and my questions have mainly been addressed.

Congratulations for this beautiful work!

The modularity and combined flexibility and rigidity of IgG3 hinges is really amazing!

We thank the reviewer for their thoughtful revision and kind words.

Minor concerns are the following:

Further attention is required for a correct use of labels in the Figure 1 legend , as well as for the one used in line 122.

Labels in Figure 1 and line 122 have been updated.

I could not see the end of the Figure 3 legend.

The legend of the figure is now visible.

I was also quite surprised by the line 320 added in the discussion, but this is discussion.

In line with your comment, we agree and have removed the added line in 320.

Reviewer #3:

We thank the review for taking the time to review our manuscript again in great detail, and for their very helpful comments.

The revised manuscript is much improved, but not all reviewer comments from the original submission are satisfactorily addressed

The methods are not completely clear, but the data in figures 2 and S6 seem not to have been aligned using independent half datasets, and the authors still use a global alignment step in Dynamo in figure S9. Limiting some degrees of freedom using the adjacent bilayer does not alter the fact that splitting the dataset into two halves for independent processing avoids many possible processing pitfalls, gives confidence in the result and allows a clear assessment of resolution. There are many possible pitfalls when aligning flexible and variable objects from small datasets with small masks. In my opinion authors should have enough confidence in the quality of the data to perform all processing using two independent half datasets.

In this context, “global” is used to describe the rotation scheme during alignment; i.e., full rotations were allowed around all axes. The datasets are indeed split into e/o and aligned to the lowpass filtered initial map completely independently, as described in the SI methods. We have changed the text in both the manuscript and supplementary materials to make it clear when and where we split the datasets, which hopefully removes this confusion.

We thank the reviewer for stimulating us to make our methods section clearer. The supplementary methods are extensive, but contain all of the information required for someone proficient at Dynamo to replicate our approach (which is based heavily on the methods paper from the authors of Dynamo, see reference 55. in the main text (Castano-Diez et al., 2012, Journal of Structural Biology) and the tutorial at https://wiki.dynamo.biozentrum.unibas.ch/w/index.php/Advanced_starters_guide). To aid understanding, we have changed the methods in the main text to give a more comprehensive overview of our subtomogram averaging routines, which we hope allays the reviewers' concern.

We have full confidence in our data and subtomogram averaging routines, which is why we have uploaded all of the particles to EMPIAR, so that they are accessible for everyone, whether they choose to copy our routine using Dynamo, or use their own subtomogram averaging procedures.

The authors still do not present data that give a numerical sense of the quality of fitting during model building, and it is difficult to assess how much more likely the presented model is as other alternative possible models. The revised text is improved but, in my opinion, it is still too difficult for the reader to assess whether there are any other qualitatively different models that may also be consistent with the data. For example, the methods section suggests that there are multiple possible arrangements of the Fab array, one of which is shown in 1f - are the other arrangements equally likely? For example, the authors state that C4b clearly fits into the density – how does the presented fit compare to the next best fit? For example, 1PK6 is fitted as a rigid body, but then 6FCZ is used to define the orientation – what does this mean? When comparing PDB to density, the authors should illustrate or describe whether the solution is unique, or if not, whether the next best

solution is also reasonable.

We have expanded the methods section to address these concerns. The model building is based on both the structural data presented herein, but also existing structural and biochemical knowledge. We now make it explicit where we have used such existing data.

*For the Fab array, we present the model that recapitulates most of the measured distances in the main text; this model is based on existing crystal structure data, where the crystal packing comprises both the 3.8 and 5.25 nm distances observed in our cryoEM map. However, we did indeed consider other models that contain some of the observed distances either in, or between, crystallographic unit cells, which we now present in the SI in **Fig. S6**. Furthermore, we discuss this in the main text, to give an understanding of the required and observed interactions of our postulated Fab array model. The salient paragraph is copied below for convenience:*

“We identified various Fab-Fab interfaces present in crystals present in the protein database (PDB) that exhibited the repeat distances observed here (Fig. S6). One of the models, with PDB code 5TDP, exhibited Fab-Fab distances of both 3.8 nm and 5.25 nm, and was used to construct a potential model of the aligned Fab domains forming a lattice perpendicular to the lipid bilayer (Figs. 1d,f). This model provided the potential interactions between adjacent heavy chains, but within the array there will also be interacting light chains required to extend the array in two dimensions. Although this interaction is not captured within 5TDP, it is present in other crystal structures (Fig. S6). The presence of all the necessary interacting partners in multiple deposited structures supports our finding that these can synergise on lipid membranes to provide the interactions required for formation of an extended Fab array, as observed here.”

For the IgG3-C1-C4b model building, we have greatly expanded the methods section and SI to give a comprehensive overview of where we have used a priori structural or biochemical knowledge to supplement our model. We have also included extra SI figures explaining where and how we used certain structures from the PDB to generate our model.

*For 1PK6 and 6FCZ (**Fig. S17**) we show that we used 6FCZ to get the correct orientation of 1PK6, and used 1PK6 for our model as the resolution of the crystal structure was higher. This orientation also agrees with known biochemical data, which we now specifically cite in the methods section.*

*For the alignment of C4b we used a 5JPM, which shows the interaction of C4 with MASP-2, a protein involved in the complement lectin pathway that is homologous to C1s (shown in **Fig. S18**), which reveals the C4b CTC and TED domains are distal and proximal to the lipid membrane, respectively. The higher resolution cryoEM map of IgM-C1 interacting with C4b (EMD-4945) also shows this same C4b orientation. We now state these considerations within the model building section.*

There is no legend for panel 1f.

We updated the legend of figure 1, thank you for bringing this to our attention.

Reviewer #4:

The authors have addressed my comments and questions adequately & therefore I recommend the updated version of the manuscript for publication.

We are happy that we could address all of their concerns and thank the review again for their impact on our work.

REVIEWERS' COMMENTS

Reviewer #3 (Remarks to the Author):

The authors have adequately addressed my concerns in this revised version, and will deposit data to allow those interested to validate the results. In my opinion, this version of the manuscript should be published.